# CHASE: Learning Convex Hull Adaptive Shift for Skeleton-based Multi-Entity Action Recognition

**Yuhang Wen**
Sun Yat-sen University
wenyh29@mail2.sysu.edu.cn

**Mengyuan Liu**[*]
State Key Laboratory of General Artificial Intelligence
Peking University, Shenzhen Graduate School
nkliuyifang@gmail.com

**Songtao Wu**
Sony R&D Center China
Songtao.Wu@sony.com

**Beichen Ding**[*]
Sun Yat-sen University
dingbch@mail.sysu.edu.cn

## Abstract

Skeleton-based multi-entity action recognition is a challenging task aiming to identify interactive actions or group activities involving multiple diverse entities. Existing models for individuals often fall short in this task due to the inherent distribution discrepancies among entity skeletons, leading to suboptimal backbone optimization. To this end, we introduce a Convex Hull Adaptive Shift based multi-Entity action recognition method (CHASE), which mitigates inter-entity distribution gaps and unbiases subsequent backbones. Specifically, CHASE comprises a learnable parameterized network and an auxiliary objective. The parameterized network achieves plausible, sample-adaptive repositioning of skeleton sequences through two key components. First, the Implicit Convex Hull Constrained Adaptive Shift ensures that the new origin of the coordinate system is within the skeleton convex hull. Second, the Coefficient Learning Block provides a lightweight parameterization of the mapping from skeleton sequences to their specific coefficients in convex combinations. Moreover, to guide the optimization of this network for discrepancy minimization, we propose the Mini-batch Pair-wise Maximum Mean Discrepancy as the additional objective. CHASE operates as a sample-adaptive normalization method to mitigate inter-entity distribution discrepancies, thereby reducing data bias and improving the subsequent classifier's multi-entity action recognition performance. Extensive experiments on six datasets, including NTU Mutual 11/26, H2O, Assembly101, Collective Activity and Volleyball, consistently verify our approach by seamlessly adapting to single-entity backbones and boosting their performance in multi-entity scenarios. Our code is publicly available at https://github.com/Necolizer/CHASE.

## 1 Introduction

Multi-entity action recognition, a challenging task derived from action recognition [1, 2, 3, 4, 5, 6, 7, 8, 9], aims to find the optimal estimator of the mapping from multi-entity motions to semantic labels, where entities involved can range from human bodies [10, 11], hands [12] to various objects [13]. Recent approaches predominantly rely on skeletal data for addressing this challenge [10, 11, 14], given that skeletons serve as a concise representation of spatiotemporal features [15, 16, 17, 18, 19, 20, 21]. This task has broad applications in human-robot interaction [22, 23], scene understanding [24, 25, 26, 27, 28], human motion analysis [29, 30, 31, 32, 33, 34], etc.

---

[*]Corresponding Authors.

38th Conference on Neural Information Processing Systems (NeurIPS 2024).

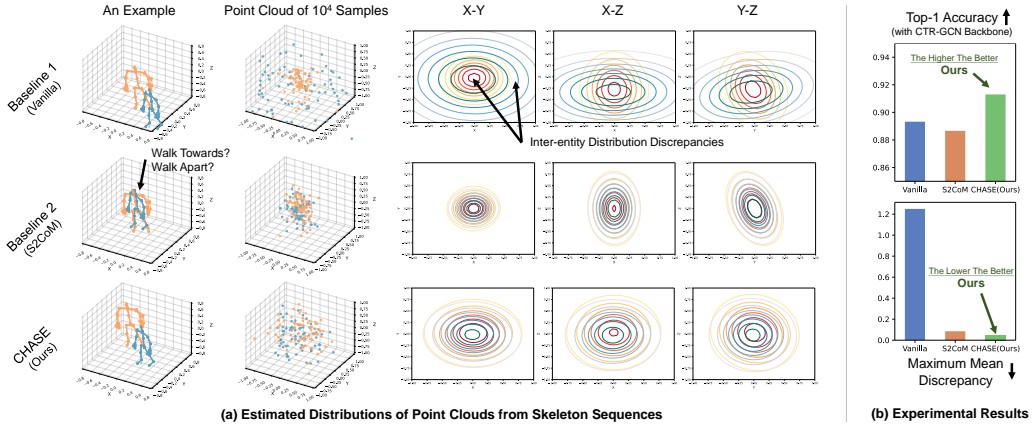

Figure 1: **Inter-entity distribution discrepancies in multi-entity action recognition task.** (a) We delineate three distinct settings: *Vanilla* (a common practice), *S2CoM* (an intuitive baseline approach), and *CHASE* (our proposed method). Column 2 illustrates spatiotemporal point clouds defined by the skeletons over $10^4$ sequences. Column 3-5 depict the projections of estimated distributions of these point clouds onto the x-y, z-x, and y-z planes. These projections reveal significant inter-entity distribution discrepancies when using *Vanilla*. (b) The discrepancies observed in *Vanilla* introduce bias into backbone models, leading to unsatisfactory performance. Although *S2CoM* can reduce these discrepancies, it makes the classifiers produce wrong predictions due to a complete loss of inter-entity information. With the lowest inter-entity discrepancy, our method unbiases the subsequent backbone to get the highest accuracy, underscoring its efficacy.

Experiments have revealed that network architectures tailored for single-entity actions get unsatisfactory performance when confronted with multi-entity actions [10, 35]. This inadequacy can be attributed to a common practice [36, 37, 38, 39, 40] observed in treating interactions: each entity is encoded independently using the same single-entity backbone, and their features are averaged for recognition. This practice is based on an empirical assumption that each entity is independent and identically distributed (i.i.d.). But we demonstrate that different entities depicted by skeletons exhibit evident non-i.i.d. characteristics. Fig. 1 (a) Row 1 reveals significant inter-entity distribution discrepancies using estimated distributions of joints from distinct entities. Such discrepancies can introduce bias into the backbone models, leading to suboptimal optimization and performance. It explains why multi-entity action modeling usually diverges from the single-entity one.

Using local coordinates for each entity holds promise in rendering them i.i.d., achieved by shifting individual origins to the per-entity spatiotemporal centers of mass (S2CoM), as depicted in Fig. 1 (a) Row 2. S2CoM is a straightforward and intuitive baseline to address this problem. However, this approach exacts a significant toll as it entails a complete loss of inter-entity information. Experimental results corroborate this notion, as illustrated in Fig. 1 (b), showcasing the detrimental impact of lacking inter-entity measurements on recognition performance. Nonetheless, this endeavor sparks an insightful realization: the potential for narrowing distribution gaps through origin shifts, thereby improving the performance of single-entity backbones in multi-entity scenarios. A natural question arises: Can we reduce the bias by finding the optimal sample-adaptive shift in $\mathbb{R}^3$ that minimizes the distribution discrepancies among entities?

To address the inter-entity distribution discrepancy problem, we propose a Convex Hull Adaptive Shift based multi-Entity action recognition method (CHASE). Serving as an additional normalization step, CHASE aims to accompany other single-entity backbones for enhanced multi-entity action recognition. Our main insight lies in the adaptive repositioning of skeleton sequences to mitigate inter-entity distribution gaps, thereby unbiasing the subsequent backbone and boosting its performance. Specifically, CHASE consists of a learnable parameterized network and an auxiliary objective. The parameterized network can achieve plausible and sample-adaptive repositioning of skeleton sequences through two crucial components. First, the Implicit Convex Hull Constrained Adaptive Shift (ICHAS) ensures that the new origin of the coordinate system is within the skeleton convex hull. Second, the Coefficient Learning Block (CLB) provides a lightweight parameterization of the mapping from skeleton sequences to their specific coefficients in ICHAS. Moreover, to guide the optimization of this network for discrepancy minimization, we propose the Mini-batch

Pair-wise Maximum Mean Discrepancy (MPMMD) as the additional objective. This loss function quantifies pair-wise entity discrepancies using maximum mean discrepancy and integrates mini-batch sampling strategies to estimate the expectation. In conclusion, CHASE works as a sample-adaptive normalization method to mitigate inter-entity distribution discrepancies, which can reduce bias in the subsequent classifier and enhance its multi-entity action recognition performance.

The contributions of this paper are three-fold:

1. To the best of our knowledge, we are the first to investigate the issue of inter-entity distribution discrepancies in multi-entity action recognition. Our proposed method, Convex Hull Adaptive Shift for Multi-Entity Actions, effectively addresses this challenge. Our main idea is adaptively repositioning skeleton sequences to mitigate inter-entity distribution gaps, thereby unbiasing the subsequent backbones and boosting their performance.

2. Serving as an additional normalization step for backbone models, CHASE consists of a learnable network and an auxiliary objective. Specifically, this network is formulated by the Implicit Convex Hull Constrained Adaptive Shift, together with the parameterization of a lightweight Coefficient Learning Block, which learns sample-adaptive origin shifts within skeleton convex hull. Additionally, the Mini-batch Pair-wise Maximum Mean Discrepancy objective is proposed to guide the discrepancy minimization.

3. Experiments on NTU Mutual 11, NTU Mutual 26, H2O, Assembly101, Collective Activity Dataset and Volleyball Dataset consistently verify our proposed method by improving performance of single-entity backbones in multi-entity action recognition task.

## 2 Related Work

### 2.1 Skeleton-based Action Recognition

**Datasets & Models**. Datasets [41, 42, 43, 44] proffering annotated or estimated skeleton sequences support the development of skeleton-based action recognition. Based on these benchmarks, a significant body of works focus on the design of artificial neural network architecture for more effective skeleton-based action recognition. Early models rely on the basic architecture of Recurrent Neural Network to capture temporal motions [45, 46, 47, 48, 49, 50]. Graph Convolution Network (GCN) shows predominated popularity as various graph convolution operators being proposed [36, 37, 38, 51, 52, 53, 54, 55, 56, 57, 58]. Recent progress of the model design is largely driven by adopting self-attention mechanism and transformer architecture [39, 40, 59, 60, 61, 62, 63, 64].

**Optimization Objectives**. Several works have explored additional optimization objectives beyond the commonly used cross-entropy (CE) loss to ensure robust recognition [37, 65], address challenging open-set problems [16], or integrate supplementary natural language descriptions [66, 67].

However, existing methods are usually developed under the empirical assumption that entities are i.i.d. allowing the backbones to learn representations of actions concerning only one entity [36, 37, 38, 39, 40, 53]. However, when confronted with multi-entity interactions, their common practice of feeding the backbone separately often proves inadequate. Our proposed approach can seamlessly adapt to these existing methods, boosting their performance by minimizing the distribution discrepancies.

### 2.2 Skeleton-based Multi-Entity Action Recognition

**Interactive Actions**. Addressing datasets featuring two-person actions [41, 42, 68, 69, 70, 71, 72] or egocentric hand-object interactions [12, 13, 73, 74] necessitates effective interaction modeling. This spurs the development of various interaction recognition models leveraging human body and hand graph priors [10, 75]. Notably, the introduction of the general interactive action recognition task [35] unifies diverse interactions across various entity types, including person-to-person [10, 11, 14, 35, 76, 77], hand-to-hand [12, 35, 78, 79] and hand-to-object [13, 35, 75, 79, 80, 81] interactions.

**Group Activities**. Another interesting area of study is group activities [82, 83, 84], which involve more entities and may include irrelevant individual motions [85, 86]. To this end, recent works usually leverage compositional reasoning from group skeletons, either alone or in combination with additional modalities, to achieve promising results [87, 88, 89, 90, 91, 92, 93, 94, 95, 96].

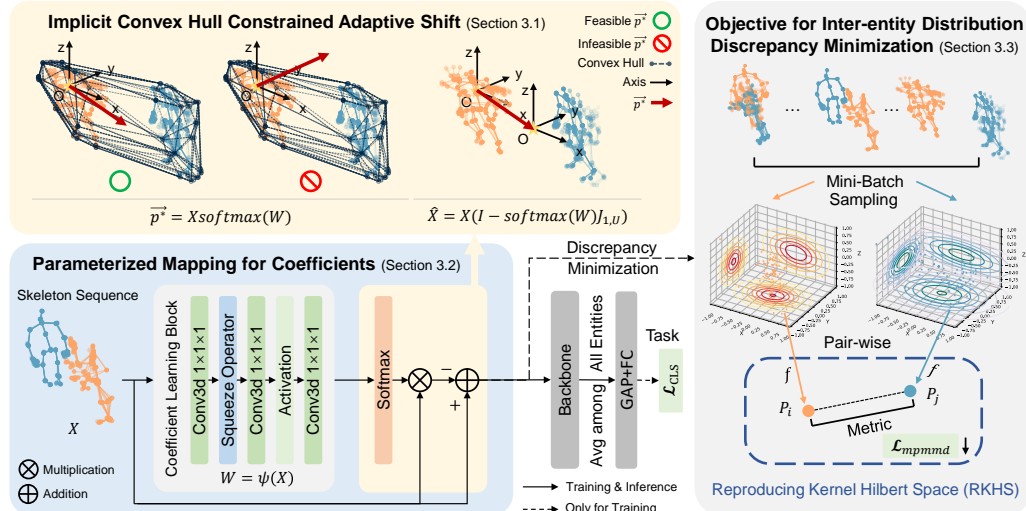

Figure 2: **The overall framework of the proposed CHASE for multi-entity action recognition.** Given a skeleton sequence of multi-entity action as input, CHASE executes an implicit convex hull constrained adaptive shift with the Coefficient Learning Block, implemented as a lightweight backbone wrapper. CHASE also collects pair-wise shifted skeletons within mini-batches, effectively alleviating inter-entity distribution discrepancies by introducing an additional objective.

While these works demonstrate satisfactory performance through interaction modelling, some may encounter model scalability issues when confronted with the factorial growth of inter-entity interactions [10, 13, 35, 75]. Moreover, they usually lack sufficient justification for why multi-entity action modeling significantly diverges from the single-entity one [10, 35, 79, 81]. In this paper, we delve into the inter-entity distribution discrepancy problem and introduce CHASE as a solution to minimize discrepancies. Through our proposed method, we aim to demonstrate that single-entity backbones can work well in multi-entity settings.

## 3    CHASE

Fig. 2 presents the framework of our proposed CHASE for skeleton-based multi-entity action recognition. We begin by presenting the formulation of the implicit convex hull constrained adaptive shift in Section 3.1, followed by the design of a lightweight Coefficient Learning Block in Section 3.2. In Section 3.3, we subsequently introduce an additional objective termed Mini-batch Pair-wise Maximum Mean Discrepancy to further mitigate inter-entity distribution discrepancies.

### 3.1    Implicit Convex Hull Constrained Adaptive Shift

The observed inter-entity distribution discrepancy in multi-entity skeleton sequences stems from the initial configuration of the world coordinate system. To mitigate this discrepancy, we propose an adaptive shift mechanism for each multi-entity skeleton sequence. It guides the origin to a sample-adaptive location, aiming to render each entity approximately i.i.d.. Moreover, based on the empirical assumption that the origin should not be far away from the skeletons, we implicitly constrain the new origin to remain within the skeleton convex hull by proving a simple but crucial proposition.

Consider a scenario where $E$ interactive entities (e.g. persons) engage in purposeful activities over a duration of $T$, and the pose of each entity is indicated by $J$ joints with $C$ Cartesian coordinates. The skeleton sequence of a multi-entity action is defined as $X \in \mathbb{R}^{C \times T \times J \times E}$. For clarity we denote $U = T \times J \times E$. Given points $\vec{p}_i \in \mathbb{R}^{C \times 1}$ in $X \in \mathbb{R}^{C \times U}$, the subtraction $\hat{\vec{p}}_i = \vec{p}_i - \vec{p^*}(1 \leq i \leq U)$ defines a shift of origin for them, where $\hat{\vec{p}}_i, \vec{p^*} \in \mathbb{R}^{C \times 1}$. This can be expressed in matrix form as:

$$\hat{X} = X - \vec{p^*}J_{1,U}, \tag{1}$$

where $J_{1,U} \in \mathbb{R}^{1 \times U}$ is a matrix of ones, and $\hat{X}$ is the shifted skeleton sequence. Now the problem is to make the shift vector $\vec{p^*}$ adaptive to $X$. A naive implementation is the linear combination:

$$\hat{X} = X - \vec{p^*}J_{1,U} = X(I - WJ_{1,U}), \tag{2}$$

where $I \in \mathbb{R}^{U \times U}$ and the weight matrix $W \in \mathbb{R}^{U \times 1}$.

However, optimizing $W$ can be challenging without constraints, as $\vec{p^*}$ could potentially be any point in $\mathbb{R}^3$. It is therefore reasonable to constrain $\vec{p^*}$ by incorporating the definition of the **Convex Hull**.

**Definition 1** (Convex Hull [97]). The convex hull $S$ of a given set $X$ can be defined as: 1) The (unique) minimal convex set containing $X$. 2) The set of all convex combinations of points in $X$. These definitions are equivalent.

We jump to the formulation of the implicit skeleton convex hull constrained adaptive shift vector by proving the following proposition:

**Proposition 1.** *The implicit skeleton convex hull constrained adaptive shift vector is formulated as*

$$\vec{p^*} = X\text{softmax}(W), \tag{3}$$

*where $X \in \mathbb{R}^{C \times U}$, $W \in \mathbb{R}^{U \times 1}$, and $\vec{p^*} \in \mathbb{R}^{C \times 1}$. $\vec{p^*}$ in Eq. 3 is an element in the set of all convex combinations of points in $X$. It is also a point that lies in the minimal convex set containing $X$.*

*Proof.* The first half of this proposition is equivalent to show that the matrix product of $X$ and $\text{softmax}(W)$ is a convex combination of $X$. $X$ is a set of points $\vec{p}_1, \ldots, \vec{p}_U$ with $C$ Cartesian coordinates. We denote $\text{softmax}(W)$ as $\tilde{W}$ with component $\tilde{\alpha}_i$, which is formulated as

$$\tilde{\alpha}_i = \frac{e^{\alpha_i}}{\sum_{j=1}^{U} e^{\alpha_j}} \quad (1 \le i \le U), \tag{4}$$

where $\alpha_i$ is a component of $W$. By applying function $\text{softmax} : \mathbb{R}^U \mapsto (0, 1)^U$, each component $\tilde{\alpha}_i$ of $\tilde{W}$ will be in the interval $(0, 1)$, and the components will add up to 1. Thus we have $\vec{p^*} = \sum_{i=1}^{U} \tilde{\alpha}_i \vec{p}_i$, where all $\tilde{\alpha}_i \in \mathbb{R}$ satisfy $\tilde{\alpha}_i > 0$ and $\sum_{i=1}^{U} \tilde{\alpha}_i = 1$. This is sufficient for the definition of a convex combination, which only requires $\tilde{\alpha}_i \ge 0$. Then the second half of this proposition is evident with the equivalence of definitions in Def. 1. $\qquad\square$

Proposition 1 also implies that all possible $\vec{p^*}$ constitute a subset $\tilde{S}$ of the convex hull $S$ defined by the skeleton joints for all entities during the action period:

$$\tilde{S} = \left\{ \sum_{i=1}^{U} \tilde{\alpha}_i \vec{p}_i \,\middle|\, \vec{p}_i \in X, \sum_{i=1}^{U} \tilde{\alpha}_i = 1, \tilde{\alpha}_i \in (0, 1) \right\} \subset S, \tag{5}$$

which specifically is the interior of $S$ (i.e., the open convex hull of $X$). We provide an example of the feasible $\vec{p^*}$ in the interior of $S$, marked by a green circle in Fig. 2. The center of mass (CoM) $\bar{\vec{p}}$ is also in the set $\tilde{S}$, proven by simply taking all $\tilde{\alpha}_i = 1/U (1 \le i \le U)$.

With Eq. 1 and Eq. 3, we introduce Implicit Convex Hull Constrained Adaptive Shift as:

$$\hat{X} = X(I - \text{softmax}(W)J_{1,U}), \tag{6}$$

where $W$ is coefficients needed to be optimized. In Eq. 2, the search space for $\vec{p^*}$ encompasses the entire $R^3$. However, in Eq. 6, it's restricted to the open convex hull $\tilde{S}$. We optimize the weights for each point under the constraint of the skeleton convex hull, subsequently deriving the adaptive shift vector for each sample. Applying a softmax function implicitly constrains $\vec{p^*}$ to remain within the convex hull $S$, while preserving inter-entity measurements. Consequently, the subtraction between the point set and the shift vector repositions the origin to a specific point in the open convex hull.

## 3.2 Parameterized Mapping for Coefficients

In this section, a lightweight Coefficient Learning Block is introduced to parameterize the mapping from the input skeleton sequence to the weight matrix. This parameterization allows CHASE to achieve sample-adaptive coefficients beyond sample-adaptive shifts formulated in Section 3.1.

In Eq. 6, we note that the first-order partial derivative of $\hat{X}$ with respect to $X$ is

$$\frac{\partial \hat{X}}{\partial X} = I - J_{U,1}\text{softmax}(W^T), \tag{7}$$

whose result is constant. This implies that the same learnt weight matrix $W$ is applied to all different $X$s when getting adaptive $\vec{p}^*$s. To make the coefficients $W$ dependent on the input $X$, a mapping $\psi : \mathbb{R}^{C \times U} \mapsto \mathbb{R}^{U \times 1}$ is expected to map $X$ to $W$.

As depicted in Fig. 2, we parameterize the nonlinear mapping $\psi$ as a sequence of learnable layers, termed the Coefficient Learning Block. This lightweight CLB can be formulated as follows:

$$W = \psi(X) = W_3\delta(W_2\phi(W_1X + b)), \tag{8}$$

where $W_1 \in \mathbb{R}^{C_1 \times C}, W_2 \in \mathbb{R}^{C_2 \times C_1}, W_3 \in \mathbb{R}^{U \times C_2}$ are weight matrices, $b$ is a bias matrix, $\phi : \mathbb{R}^{C_1 \times U} \mapsto \mathbb{R}^{C_1 \times 1}$ is a squeeze operator [98] and $\delta$ is an activation function. Using a dimensionality-reduction layer and a dimensionality-increasing layer around the non-linearity is a common gating mechanism parameterization [98, 99]. Hence, we ensure $U \geq C_1 > C_2$.

## 3.3 Objective for Inter-entity Distribution Discrepancy Minimization

To facilitate CHASE optimization, we introduce an additional objective aimed at minimizing the inter-entity distribution discrepancies of the shifted skeleton sequences. This objective quantifies the pair-wise discrepancies and employs mini-batch sampling strategies to estimate the expectation.

Maximum mean discrepancy is a metric used to measure the distance between distributions, defined as the distance between their embeddings in the reproducing kernel Hilbert space (RKHS) $\mathcal{H}$:

$$\text{MMD}(P, Q) = \sup_{\|f\|_{\mathcal{H}} \leq 1} \left( \mathbb{E}[f(x)] - \mathbb{E}[f(y)] \right), \tag{9}$$

where $\sup(\cdot)$ denotes the supremum. It is equivalent to finding the RKHS function $f$ that maximizes the difference in expectations between the two probability distributions $P(x)$ and $Q(y)$.

Suppose each entity distribution is denoted as $P^i(1 \leq i \leq E)$ for $E$ entities, we measure the distance of all pair-wise distributions using the empirical mean

$$\mathbb{E}_{r(z)}[\text{MMD}(z)] = \sum_{i=1}^{E-1} \sum_{j=i+1}^{E} \text{MMD}(P^i, P^j)/\text{C}(E, 2), \tag{10}$$

where $z = (P^i, P^j)(1 \leq i, j \leq E, i \neq j)$ with the probability density $r(z)$, and $C(E, 2)$ denotes a combination of $E$ things taken 2 at a time without repetition. We adopt two approximations for computational efficiency. The first involves estimating $\mathbb{E}[f(x)]$ in Eq. 9 using a mini-batch of $x$. The second approximation concerns the right-hand side of Eq. 10, which is impractical due to its complexity of $O(n!)$ in terms of the entity count. Instead, it can be approximated by uniformly sampling a mini-batch of $M$ entity pairs from all possible C$(E, 2)$ combinations $z$:

$$\mathbb{E}_{r(z)}[\text{MMD}(z)] \approx \frac{1}{M}\sum_{m=1}^{M} \text{MMD}(z_m). \tag{11}$$

We denote Eq. 11 with the above two approximations to be the Mini-batch Pair-wise Maximum Mean Discrepancy Loss $\mathcal{L}_{mpmmd}$, thereby we have the total loss function for training:

$$\mathcal{L} = \mathcal{L}_{CLS} + \lambda\mathcal{L}_{mpmmd}, \tag{12}$$

where $\mathcal{L}_{CLS}$ is the classification loss and $\lambda$ is the trade-off weight factor for $\mathcal{L}_{mpmmd}$.

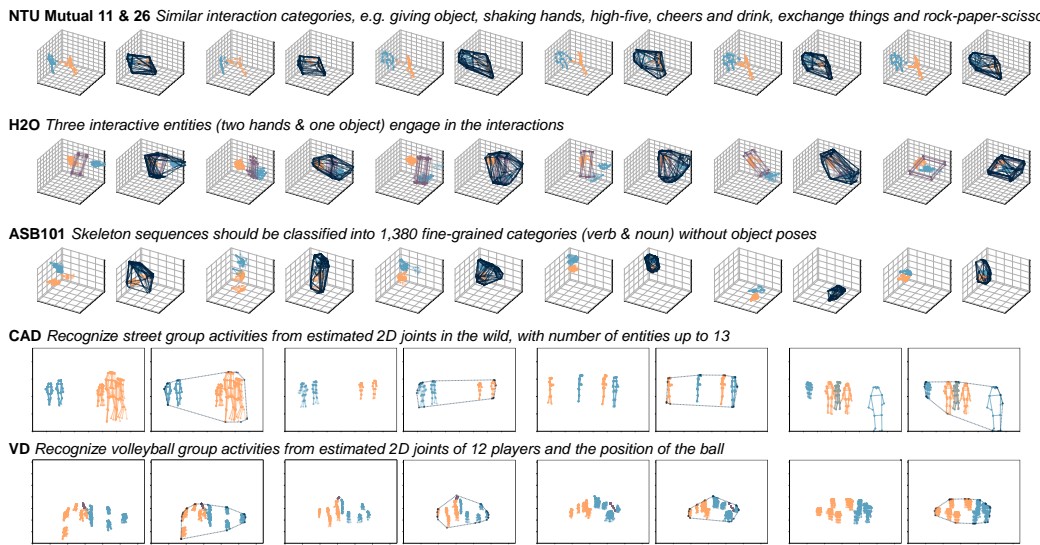

**NTU Mutual 11 & 26** *Similar interaction categories, e.g. giving object, shaking hands, high-five, cheers and drink, exchange things and rock-paper-scissors*

**H2O** *Three interactive entities (two hands & one object) engage in the interactions*

**ASB101** *Skeleton sequences should be classified into 1,380 fine-grained categories (verb & noun) without object poses*

**CAD** *Recognize street group activities from estimated 2D joints in the wild, with number of entities up to 13*

**VD** *Recognize volleyball group activities from estimated 2D joints of 12 players and the position of the ball*

Figure 3: **Visualizations of multi-entity action samples and their skeleton convex hulls.**

## 4 Experiments

### 4.1 Datasets & Settings

We conduct experiments on six multi-entity action recognition datasets. Fig. 3 presents skeletal samples in these datasets and their skeleton convex hulls, showcasing their difficulties.

**NTU Mutual 11** and **NTU Mutual 26**, respectively subsets of **NTU RGB+D** [41] and **NTU RGB+D 120** [42], consist of a variety of inter-person mutual actions. NTU Mutual 11 adopts the widely-used X-Sub and X-View criteria, while NTU Mutual 26 follows the X-Sub and X-Set criteria.

**H2O** [13] proffers 3D poses of human hands and bounding boxes of the manipulated objects, facilitating both hand-to-hand and hand-to-object interactions learning. We follow the training, validation, and test splits outlined in [13] in our experiments.

**Assembly101 (ASB101)** [12] is a large and challenging 3D manual procedural activity dataset, with 1,380 categories of interactive actions. We follow the training, validation, and test splits described in [12] for evaluations. Fine-grained actions (verb & noun) are adopted as labels in experiments.

**Collective Activity Dataset (CAD)** [85] captures people and their behaviors in public using street cameras, categorizing pedestrian collective activities into 4 groups. We adopt the same categories, individual labels, train-test split in [95]. Only 2D joint coordinates are used in our experiments.

**Volleyball Dataset (VD)** [86] consists of video clips from volleyball tournaments and includes 8 group activity classes based on volleyball terminology. We follow the Original split described in [95] for evaluation. Only estimated 2D joint coordinates are used as input features.

**Settings.** Experiments are conducted on the GeForce RTX 3070 GPUs with PyTorch. CTR-GCN [36], InfoGCN [37], STSA-Net [40] and HD-GCN [38] are chosen as our baseline models. To ensure fair comparisons, we adopt single intra-skeleton modality without multi-modality fusion following [35]. For CTR-GCN in NTU Mutual 26, we adopt input shape $X \in \mathbb{R}^{3 \times 64 \times 25 \times 2}$, segment size $(1, 1, 1)$ and $\lambda = 0.1$ in CHASE. SGD optimizer is used with Nesterov momentum of 0.9, a initial learning rate of 0.1 and a decay rate 0.1 at the 80th and 100th epoch. Batch size is set to 64. More detailed configurations for each model are provided in the Appendix.

### 4.2 Experimental Results

Table 1 shows the experimental results on different benchmarks, reporting the averaged top-1 accuracy and its standard deviation in runs with several seed initializations. We compare CHASE with vanilla counterparts (light red background) and the state-of-the-art multi-entity action recognition methods (light yellow background). By adopting our proposed CHASE, we can boost the vanilla counterparts'

Table 1: Comparisons with Skeleton-based Methods on Multi-Entity Action Datasets

| Method | Venue | NTU Mutual 26(%) | | NTU Mutual 11(%) | |
|---|---|---|---|---|---|
| | | X-Sub | X-Set | X-Sub | X-View |
| GDCN [11] | TPAMI'23 | 85.80 | 92.10 | - | - |
| SkeleTR [76] | ICCV'23 | 87.80 | 88.30 | 94.80 | 97.70 |
| ISTA-Net [35] | IROS'23 | $90.56_{(\pm0.08)}$ | $91.72_{(\pm0.30)}$ | - | - |
| AHNet-Large [83] | PR'24 | 86.43 | 86.64 | 90.85 | 93.38 |
| me-GCN [77] | arXiv'24 | 90.00 | 90.00 | 95.50 | 98.20 |
| CTR-GCN [36] | ICCV'21 | $89.32_{(\pm0.06)}$ | $90.19_{(\pm0.17)}$ | $95.94_{(\pm0.36)}$ | $98.32_{(\pm0.29)}$ |
| **+ CHASE (Ours)** | - | $\mathbf{91.30}^{\uparrow1.98}_{(\pm0.22)}$ | $\mathbf{92.34}^{\uparrow2.15}_{(\pm0.10)}$ | $\mathbf{96.45}^{\uparrow0.51}_{(\pm0.05)}$ | $\mathbf{98.83}^{\uparrow0.51}_{(\pm0.13)}$ |
| InfoGCN [37](k=1) | CVPR'22 | $90.22_{(\pm0.13)}$ | $91.13_{(\pm0.16)}$ | $95.51_{(\pm0.10)}$ | $97.76_{(\pm0.22)}$ |
| **+ CHASE (Ours)** | - | $\mathbf{91.86}^{\uparrow1.64}_{(\pm0.05)}$ | $\mathbf{92.41}^{\uparrow1.28}_{(\pm0.34)}$ | $\mathbf{96.35}^{\uparrow0.84}_{(\pm0.18)}$ | $\mathbf{98.25}^{\uparrow0.49}_{(\pm0.25)}$ |
| STSA-Net [40] | Neuro.'23 | $88.41_{(\pm0.01)}$ | $90.19_{(\pm0.11)}$ | $95.96_{(\pm0.09)}$ | $98.47_{(\pm0.09)}$ |
| **+ CHASE (Ours)** | - | $\mathbf{89.77}^{\uparrow1.36}_{(\pm0.18)}$ | $\mathbf{91.54}^{\uparrow1.35}_{(\pm0.12)}$ | $\mathbf{96.63}^{\uparrow0.68}_{(\pm0.10)}$ | $\mathbf{98.73}^{\uparrow0.26}_{(\pm0.08)}$ |
| HD-GCN [38](CoM=1) | ICCV'23 | $88.25_{(\pm0.44)}$ | $90.08_{(\pm0.12)}$ | $95.58_{(\pm0.10)}$ | $97.93_{(\pm0.07)}$ |
| **+ CHASE (Ours)** | - | $\mathbf{90.81}^{\uparrow2.56}_{(\pm0.13)}$ | $\mathbf{92.06}^{\uparrow1.97}_{(\pm0.21)}$ | $\mathbf{96.22}^{\uparrow0.64}_{(\pm0.05)}$ | $\mathbf{98.31}^{\uparrow0.38}_{(\pm0.07)}$ |

| Method | Venue | H2O(%) | ASB101(%) | CAD(%) | VD(%) |
|---|---|---|---|---|---|
| AT [26] | CVPR'20 | - | - | - | 92.30 |
| ISTA-Net [35] | IROS'23 | $89.09_{(\pm1.21)}$ | $28.01_{(\pm0.06)}$ | $87.16_{(\pm2.55)}$ | $91.40_{(\pm0.23)}$ |
| H2OTR [80] | CVPR'23 | 90.90 | - | - | - |
| EffHandEgoNet [81] | arXiv'24 | 91.32 | - | - | - |
| AHNet-Large [83] | PR'24 | - | - | 89.32 | 84.31 |
| CTR-GCN [36] | ICCV'21 | $81.68_{(\pm0.85)}$ | $27.83_{(\pm0.45)}$ | $80.45_{(\pm2.29)}$ | $92.66_{(\pm0.21)}$ |
| **+ CHASE (Ours)** | - | $\mathbf{91.05}^{\uparrow9.37}_{(\pm1.98)}$ | $\mathbf{28.03}^{\uparrow0.21}_{(\pm0.30)}$ | $\mathbf{89.61}^{\uparrow9.16}_{(\pm0.20)}$ | $\mathbf{92.89}^{\uparrow0.24}_{(\pm0.15)}$ |
| InfoGCN [37](k=1) | CVPR'22 | $76.24_{(\pm3.93)}$ | $27.18_{(\pm0.10)}$ | $83.07_{(\pm0.46)}$ | $91.77_{(\pm0.15)}$ |
| **+ CHASE (Ours)** | - | $\mathbf{83.47}^{\uparrow7.23}_{(\pm2.89)}$ | $\mathbf{27.36}^{\uparrow0.18}_{(\pm0.12)}$ | $\mathbf{84.18}^{\uparrow1.11}_{(\pm2.91)}$ | $\mathbf{92.00}^{\uparrow0.23}_{(\pm0.15)}$ |
| STSA-Net [40] | Neuro.'23 | $92.29_{(\pm0.52)}$ | $27.70_{(\pm0.19)}$ | $80.20_{(\pm3.60)}$ | $92.52_{(\pm0.52)}$ |
| **+ CHASE (Ours)** | - | $\mathbf{94.77}^{\uparrow2.48}_{(\pm1.36)}$ | $\mathbf{27.81}^{\uparrow0.11}_{(\pm0.13)}$ | $\mathbf{85.93}^{\uparrow5.73}_{(\pm2.46)}$ | $\mathbf{92.78}^{\uparrow0.26}_{(\pm0.41)}$ |
| HD-GCN [38](CoM=1) | ICCV'23 | $72.73_{(\pm0.41)}$ | $27.31_{(\pm0.36)}$ | $76.93_{(\pm4.38)}$ | $91.32_{(\pm0.02)}$ |
| **+ CHASE (Ours)** | - | $\mathbf{81.61}^{\uparrow8.88}_{(\pm1.03)}$ | $\mathbf{27.50}^{\uparrow0.19}_{(\pm0.24)}$ | $\mathbf{82.39}^{\uparrow5.46}_{(\pm1.61)}$ | $\mathbf{92.00}^{\uparrow0.68}_{(\pm0.07)}$ |

performance by a noticeable margin in most settings. It yields varying degrees of accuracy improvement across different baseline models and benchmarks, owing to differences in model parameter count, training objective, data scale, etc. Compared to models with complicated interaction designs, CHASE can help single action backbones achieve the state-of-the-art performance in interaction recognition by outperforming ISTA-Net [35], AHNet-Large [83], etc. In group activities recognition task, which is more challenging for single-entity backbones, CHASE can help achieve competitive performance. Fig. 4 visualizes that CHASE can effectively alleviate the potential inter-entity distribution discrepancies across a range of data scales, thereby ensuring robust backbone optimization and inference. UMAP [100] visualization in Fig. 5 demonstrates our proposed CHASE differentiate similar multi-entity actions better by assisting backbones to learn more distinctive representations.

### 4.3 Ablation Study

In this section, we conduct ablation studies on the widely-adopted benchmarks NTU Mutual 26 and NTU Mutual 11 with only joint modality.

**Comparison with other alternatives**. We compare our proposed CHASE with several alternatives as follows: 1) Vanilla: Use the raw world coordinates or pixel coordinates. 2) S2CoM: Shift

Table 2: Comparison with Other Alternatives

| Method | Acc (%) | $\Delta$ (%) |
|---|---|---|
| Vanilla | $89.32_{(\pm0.06)}$ | - |
| S2CoM | $88.66_{(\pm0.26)}$ | $-0.67$ |
| BatchNorm | $89.06_{(\pm0.16)}$ | $-0.27$ |
| ER [35] | $89.34_{(\pm0.15)}$ | $+0.02$ |
| Aug | $89.72_{(\pm0.04)}$ | $+0.40$ |
| S2CoM†/STD | $90.29_{(\pm0.06)}$ | $+0.97$ |
| S2CoM† | $90.79_{(\pm0.10)}$ | $+1.47$ |
| **CHASE (Ours)** | $\mathbf{91.30}_{(\pm0.22)}$ | $\mathbf{+1.98}$ |

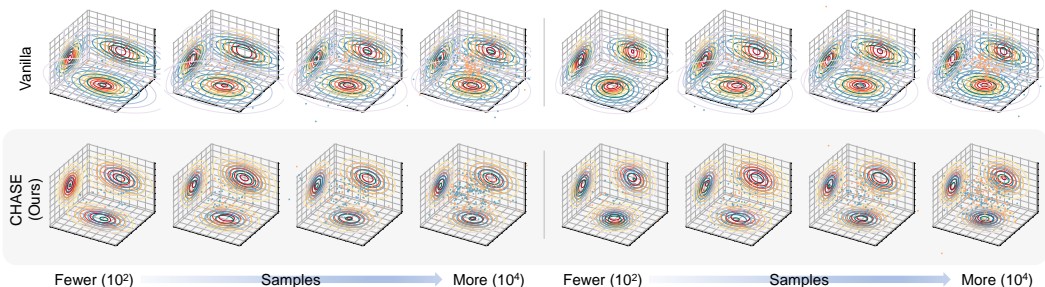

Figure 4: **Qualitative results of CHASE**. Different entity distributions are denoted by blue and orange. CHASE effectively mitigates inter-entity distribution discrepancies, demonstrating its clear effectiveness across a range of data scales, from small to large.

Table 3: Analysis of Inter-entity Distribution Discrepancies

| Set | Method | Avg KLD ↓ | JSD ↓ | BD ↓ | HD ↓ | MMD ↓ |
|---|---|---|---|---|---|---|
| I | Vanilla | $1.07_{(\pm0.25)}$ | $0.19_{(\pm0.04)}$ | $0.25_{(\pm0.06)}$ | $0.46_{(\pm0.06)}$ | $0.94_{(\pm0.54)}$ |
| | **CHASE (Ours)** | $\mathbf{0.39}_{(\pm0.09)}$ | $\mathbf{0.08}_{(\pm0.02)}$ | $\mathbf{0.10}_{(\pm0.02)}$ | $\mathbf{0.30}_{(\pm0.03)}$ | $\mathbf{0.05}_{(\pm0.02)}$ |
| II | Vanilla | $1.00_{(\pm0.23)}$ | $0.18_{(\pm0.04)}$ | $0.23_{(\pm0.05)}$ | $0.45_{(\pm0.05)}$ | $1.03_{(\pm0.60)}$ |
| | **CHASE (Ours)** | $\mathbf{0.45}_{(\pm0.08)}$ | $\mathbf{0.10}_{(\pm0.02)}$ | $\mathbf{0.11}_{(\pm0.02)}$ | $\mathbf{0.32}_{(\pm0.03)}$ | $\mathbf{0.07}_{(\pm0.02)}$ |
| III | Vanilla | $0.72_{(\pm0.14)}$ | $0.14_{(\pm0.02)}$ | $0.17_{(\pm0.03)}$ | $0.39_{(\pm0.04)}$ | $1.25_{(\pm0.60)}$ |
| | **CHASE (Ours)** | $\mathbf{0.41}_{(\pm0.08)}$ | $\mathbf{0.08}_{(\pm0.02)}$ | $\mathbf{0.10}_{(\pm0.02)}$ | $\mathbf{0.30}_{(\pm0.03)}$ | $\mathbf{0.05}_{(\pm0.04)}$ |
| IV | Vanilla | $0.75_{(\pm0.14)}$ | $0.14_{(\pm0.03)}$ | $0.17_{(\pm0.03)}$ | $0.40_{(\pm0.04)}$ | $1.15_{(\pm0.56)}$ |
| | **CHASE (Ours)** | $\mathbf{0.41}_{(\pm0.07)}$ | $\mathbf{0.08}_{(\pm0.01)}$ | $\mathbf{0.09}_{(\pm0.02)}$ | $\mathbf{0.30}_{(\pm0.03)}$ | $\mathbf{0.04}_{(\pm0.03)}$ |

the individual origins to the spatiotemporal centers of mass for each entity. 3) BatchNorm: Apply an additional BatchNorm operation immediately when batches of samples are fed into the model. 4) ER (Entity Rearrangement [35]): A technique aims to eliminate the orderliness of entities for interaction modelling. 5) Aug: Apply an additional data augmentation by randomly shifting the skeleton sequences. 6) S2CoM†: Shift the origin to the spatiotemporal center of mass. 7) S2CoM†/STD: Scale according to the channel-wise standard deviations after applying S2CoM†. Results in Table 2 indicate that CHASE can outperform these alternatives by bringing the largest accuracy improvement to the vanilla CTR-GCN.

**Analysis of inter-entity distribution discrepancies.** Table 3 presents metrics evaluating the inter-entity distribution discrepancies on test sets, including Averaged Kullback-Leibler Divergence (Avg KLD), Jensen-Shannon Divergence (JSD), Bhattacharyya Distance (BD), Hellinger Distance (HD) and MMD. We measure the pair-wise distributions of sampled data points from different entities in test sets of NTU Mutual 11 X-Sub (I), X-View (II) and NTU Mutual 26 X-Sub (III), X-Set (IV). Table 3 demonstrates that CHASE significantly minimizes discrepancies across all evaluation metrics, thereby benefiting backbone learning for each entity in multi-entity actions.

**Analysis of Key Components**. We validate the effectiveness of each key component in Table 4. When removing the skeleton convex hull constraint (CHC), there is a significant drop in accuracy, exceeding 60%, for initial learning rates (lr) of 0.1 and 0.01. This substantial decline highlights the importance of CHC as a critical constraint for learning the adaptive shift. Additionally, replacing Adaptive Shift (AS) with $\hat{X} = XWJ_{1,U}$ results in a dramatic decrease in accu-

Table 4: Analysis of Key Components in CHASE

| ICHAS | | CLB | MPMMD | lr | Acc (%) | Δ (%) |
|---|---|---|---|---|---|---|
| AS | CHC | | | | | |
| ✓ | ✓ | ✓ | ✓ | 0.1 | $\mathbf{91.30}_{(\pm0.22)}$ | - |
| ✓ | | ✓ | ✓ | 0.1 | $22.65_{(\pm0.35)}$ | $-68.65$ |
| ✓ | | ✓ | ✓ | 0.01 | $86.99_{(\pm0.16)}$ | $-4.32$ |
| ✓ | ✓ | | ✓ | 0.1 | $91.20_{(\pm0.13)}$ | $-0.10$ |
| ✓ | | | ✓ | 0.1 | $22.75_{(\pm0.12)}$ | $-68.56$ |
| ✓ | | | ✓ | 0.01 | $23.51_{(\pm0.38)}$ | $-67.79$ |
| | ✓ | ✓ | ✓ | 0.1 | $20.42_{(\pm0.09)}$ | $-70.88$ |
| ✓ | ✓ | ✓ | | 0.1 | $91.17_{(\pm0.18)}$ | $-0.13$ |
| | | | | 0.1 | $89.50_{(\pm0.14)}$ | $-1.81$ |

racy, indicating that simply adding an equivalent number of trainable parameters without an adaptive shift formulation is ineffective. Table 4 further shows CHASE also benefits from CLB and MPMMD.

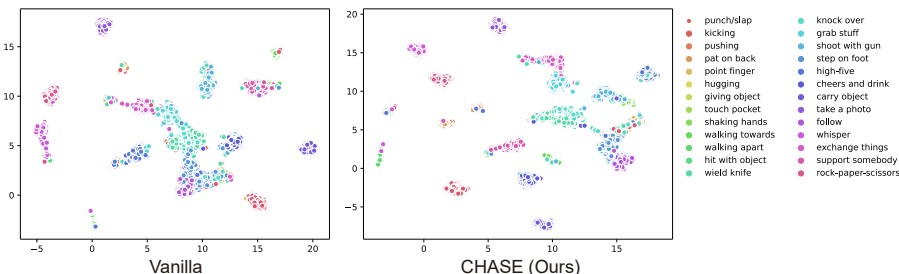

Figure 5: **UMAP [100] visualizations of multi-entity skeleton sequence representations on the test split of NTU Mutual 26 X-Sub**. Compared with Vanilla, our proposed CHASE differentiate similar multi-entity actions better by assisting backbones to learn more distinctive representations.

Table 5: Mixed Recognition on NTU RGB+D 120

| Method | X-Sub (%) | X-Set (%) |
|---|---|---|
| CTR-GCN [36] | $84.95_{(\pm 0.05)}$ | $86.90_{(\pm 0.03)}$ |
| **+ CHASE** | $\mathbf{85.36}_{(\pm 0.05)}$ | $\mathbf{86.95}_{(\pm 0.10)}$ |

**Evaluations on Mixed Recognition of Single- & Multi-Entity Actions**. Table 5 shows a 0.41% improvement in X-Sub accuracy on the entire NTU RGB+D 120. This implies that although CHASE is proposed for multi-entity actions, it is also effective in mixed recognition settings.

**Analysis of Efficiency**. As illustrated in Table 6, the number of trainable parameters of CHASE in NTU Mutual 26 configurations is about 26.37 k, resulting in a mere 1%-2% parameter increase. For computational complexity, FLOPs of CHASE is approximately 2.50 M. These metrics demonstrate that CHASE is both efficient and lightweight.

Table 6: CHASE Trainable Parameters

| Method | # Param. (M) |
|---|---|
| CTR-GCN [36] | 1.44 |
| **+ CHASE** | $1.46^{\uparrow 1.83\%}$ |
| STSA-Net [40] | 4.13 |
| **+ CHASE** | $4.16^{\uparrow 0.60\%}$ |

## 5 Conclusion

This paper proposes the Convex Hull Adaptive Shift for Multi-Entity Action Recognition (CHASE) to address the inter-entity distribution discrepancies. To the best of our knowledge, we are the first to investigate this problem and leverage discrepancy minimization to unbias the classifiers. Our approach can seamlessly adapt to existing backbone architectures and demonstrate performance improvements across six multi-entity action recognition datasets.

## Acknowledgments and Disclosure of Funding

This work was supported by Natural Science Foundation of Shenzhen (No. JCYJ20230807120801002) and National Natural Science Foundation of China (No. 62203476, No. 52105079).

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

# Appendix

The appendix is organized as follows:

# A  Supplementary Analysis of CHASE

## A.1  Preliminaries

Here we clarify some crucial definitions in the skeleton-based multi-entity action recognition task as follows.

**Definition 2** (Skeleton Sequence of A Multi-Entity Action). Suppose that $E$ interactive entities (e.g. persons) engage in a purposeful act during a period of time $T$, and the pose of each entity is indicated by $J$ joints with $C$ Cartesian coordinates. We can define the skeleton sequence of a multi-entity action as $X \in \mathbb{R}^{C \times T \times J \times E}$.

**Definition 3** (Skeleton-based Multi-Entity Action Recognition). We define the task as finding the optimal estimator $\mathcal{E}_\theta$ of the mapping $\mathcal{E} : X \mapsto Y$, where $X$ is the skeleton sequence of a multi-entity action and $Y$ is its corresponding label.

**Definition 4** (Joints & Bones). A joint within a skeleton is a point in the Cartesian coordinate system, which can also be viewed as a vector $\vec{p}_i (1 \leq i \leq J)$. A bone within a skeleton is a differential vector of two joints $\vec{p}_i$ and $\vec{p}_j$ ($1 \leq i, j \leq J$), if and only if the two joints are connected or the same one according to a prior graph (e.g. the bone connection of the human body).

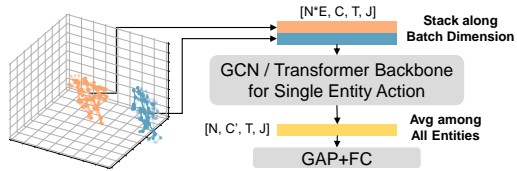

Figure 6: **The common practice in single-entity action recognition models to recognize multi-entity actions.** This late fusion strategy is used in many recent works [36, 37, 38, 39, 40].

We illustrate the common practice [36, 37, 38, 39, 40] when single-entity action recognition models meet multiple entities using Fig. 6. With vanilla world coordinates as input, they usually concatenate each entity along batch dimension, and extract high-dimensional features of each entity separately using GCN or transformer backbone. Subsequently, the individual features get averaged before undergoing global average pooling and full connection layers. Notably, this common practice is based on an empirical assumption that each entity is independent and identically distributed.

## A.2  Adaptive Shift Analysis Under Bone Representation

Most existing research uses Def. 4 for the bone modality definition [36, 53, 54, 56]. In this context, the learned adaptive shift is ineffective because shifting the origin does not affect the Euclidean distance between any two points. However, the adaptive shift could still prove effective if bones are defined differently.

Table 7: Statistics of Multi-Entity Action Recognition Datasets

| Datasets | Annotation | | | #Actions | #Joints | #Clips | #Valid Frames | #Entities | #Participants |
|---|---|---|---|---|---|---|---|---|---|
| | Body | Hand | Object | | | | | | |
| NTU Mutual 11 [41] | ✓ | | | 11 | 25 | 10,347 | 69.18 | 2.00 | 40 |
| NTU Mutual 26 [42] | ✓ | | | 26 | 25 | 24,732 | 59.36 | 2.00 | 106 |
| H2O [13] | | ✓ | ✓ | 36 | 21 | 933 | 97.29 | 3.00 | 4 |
| Assembly101 [12] | | ✓ | | 1,380 | 21 | 85,252 | 105.91 | 2.00 | 53 |
| CAD [85] | ✓ | | | 4 | 17 | 2,511 | 10.00 | 5.22 | - |
| VD [86] | ✓ | | ✓ | 8 | 17 | 4,830 | 10.00 | 13.00 | - |

**Definition 5** (k-hop Bones [37]). We denote $X_t$ as the joints at the moment $t$ in a skeleton sequence. The k-hop Bone $\tilde{X}_t^{(k)}$ at moment $t$ can be formulated as

$$\tilde{X}_t^{(k)} = (I - P^k)X_t, \tag{13}$$

where $k \geq 1$, and $P \in \mathbb{R}^{J \times J}$ is a binary adjacency matrix of a directed graph without bi-directional edges.

In the context of human body skeletons, Def. 5 uses the joint $j_r$, representing *the center of the spine*, as the root to construct the directed graph. If $k = max_v d(v) + 1$, where $max_v d(v)$ means the max hop of all joints to the root, then the k-hop Bone $\tilde{X}_t^{(k)}$ aligns with the joint definition in Def. 4. However, when $k = 1$, this equivalence doesn't necessarily hold for the bone definition in Def. 4. The reason is that some joints (such as the root) may have no in-degree, causing them to retain their original joint coordinates in Def. 5. Therefore, the k-hop bones may still contain joint information that can be modified through adaptive shift.

### A.3 Analysis of Gradient

Consider the expectation $\mathbb{E}_{r(z)}[g_\theta(z)]$ in Eq. 11, where $g_\theta$ denotes the composition of MMD (Eq. 9) and ICHAS (Eq. 6) with CLB (Eq. 8). We assume that the gradient of $g_\theta$ with respect to the parameters $\theta$ exists. Since it adopts uniform sampling, we note that the discrete probability density function $r(z)$ of $z$ is independent of the parameters $\theta$. Thus we have

$$\begin{aligned}
\nabla_\theta \mathbb{E}_{r(z)}[g_\theta(z)] &= \nabla_\theta [\sum_z r(z) g_\theta(z)] \\
&= \sum_z r(z)[\nabla_\theta g_\theta(z)] \\
&= \mathbb{E}_{r(z)}[\nabla_\theta g_\theta(z)],
\end{aligned} \tag{14}$$

which indicates that the gradient of the expectation $\nabla_\theta \mathbb{E}_{r(z)}[g_\theta(z)]$ is equivalent to the expectation of the gradient $\mathbb{E}_{r(z)}[\nabla_\theta g_\theta(z)]$. The latter can be approximated using Monte Carlo methods.

## B  Code for CHASE

CHASE is implemented as a wrapper for various skeleton-based action backbones, as illustrated in Algorithm 1. Line 18 indicates input of batch size $N$, channel $C$, number of frames $T$, number of joints $V$, number of entity $M$. Line 19 represents Eq. 8, which is to map $X$ to $W$ with function $\psi$. Line 21 represents the formulation of $\vec{p^*}$ in Eq. 3. Line 25 indicate the subtraction in Eq. 1. Line 26-27 are mini-batch sampling strategy for the Mini-batch Pair-wise Maximum Mean Discrepancy Loss. Line 28 represents the single-entity backbone. Our code is publicly available at `https://github.com/Necolizer/CHASE` with MIT license.

## C  Details of Multi-Entity Action Recognition Datasets

We conduct experiments on a range of datasets, including **NTU Mutual 11** (a subset of **NTU RGB+D** [41]), **NTU Mutual 26** (a subset of **NTU RGB+D 120** [42]), **H2O** [13], **Assembly101 (ASB101)** [12], **Collective Activity Dataset (CAD)** [85], and **Volleyball Dataset (VD)** [86]. Table 7

**Algorithm 1** CHASE Wrapper: PyTorch-like Pseudo Code

```
 1:  class CHASEWrapper(nn.Module):
 2:    def __init__(self, backbone, in_channels, num_frame, num_point, pooling_seg, num_entity,
       c1, c2):
 3:      super(CHASEWrapper, self).__init__()

 4:      out_channel = num_frame * num_point * num_entity
 5:      self.pooling_seg = pooling_seg
 6:      self.seg = self.pooling_seg[0]*self.pooling_seg[1]*self.pooling_seg[2]
 7:      self.seg_num_list = [(num_frame//self.pooling_seg[0]), (num_point//self.pooling_seg[1]),
       (num_entity//self.pooling_seg[2])]
 8:      self.seg_num = self.seg_num_list[0] * self.seg_num_list[1] * self.seg_num_list[2]
 9:      self.shift = nn.Sequential(
10:        nn.Conv3d(in_channels=in_channels, out_channels=c1, kernel_size=1),
11:        nn.AdaptiveAvgPool3d((self.pooling_seg[0], self.pooling_seg[1], self.pooling_seg[2])),
12:        nn.Conv3d(c1, c2, 1, bias=False),
13:        nn.ReLU(inplace=True),
14:        nn.Conv3d(c2, out_channel, 1, bias=False),
15:      )
16:      self.backbone = backbone

17:    def forward(self, x):
18:      N, C, T, V, M = x.size()
19:      sf = self.shift(x).view(N, T*M*V, -1)
20:      tx = rearrange(x, 'n c t v m -> n c (t v m)', t=T, m=M, v=V).contiguous()
21:      sf = (tx @ sf.softmax(dim=1)).unsqueeze(-1).expand(-1, -1, self.seg, self.seg_num)
22:      sf = rearrange(sf, 'n c (T V M) (t v m) -> n c (T t) (V v) (M m)',
23:          T=self.pooling_seg[0], V=self.pooling_seg[1], M=self.pooling_seg[2],
24:          t=self.seg_num_list[0], v=self.seg_num_list[1], m=self.seg_num_list[2]).contiguous()
25:      x = x - sf
26:      if self.training:
27:        pairs = MiniBatchSampling(x)
28:      out = self.backbone(x)

29:      if self.training:
30:        return out, pairs
31:      else:
32:        return out
```

provides details about each dataset, including their annotation types, numbers of action categories, numbers of joints, numbers of clips (samples), averaged counts of valid frames, counts of entities engaging in a multi-entity action, and numbers of participants in data collection. For CAD [85] and VD [86], which both capture videos in the wild with a variety of individuals, it is difficult to determine the exact number of participants.

# D Evaluation Metrics

In this section, we provide the detailed formulation for metrics in ablation study. Given two discrete probability distributions $P$ and $Q$ defined on the same sample space $\mathcal{X}$, we can define the following metrics:

**Averaged Kullback-Leibler Divergence (Avg KLD):**

$$
\begin{aligned}
Avg\ D_{KL} &= \frac{1}{2}[D_{KL}(P\|Q) + D_{KL}(Q\|P)] \\
&= \frac{1}{2}[\sum_{x\in\mathcal{X}} P(x)\log(\frac{P(x)}{Q(x)}) + \sum_{x\in\mathcal{X}} Q(x)\log(\frac{Q(x)}{P(x)})].
\end{aligned} \tag{15}
$$

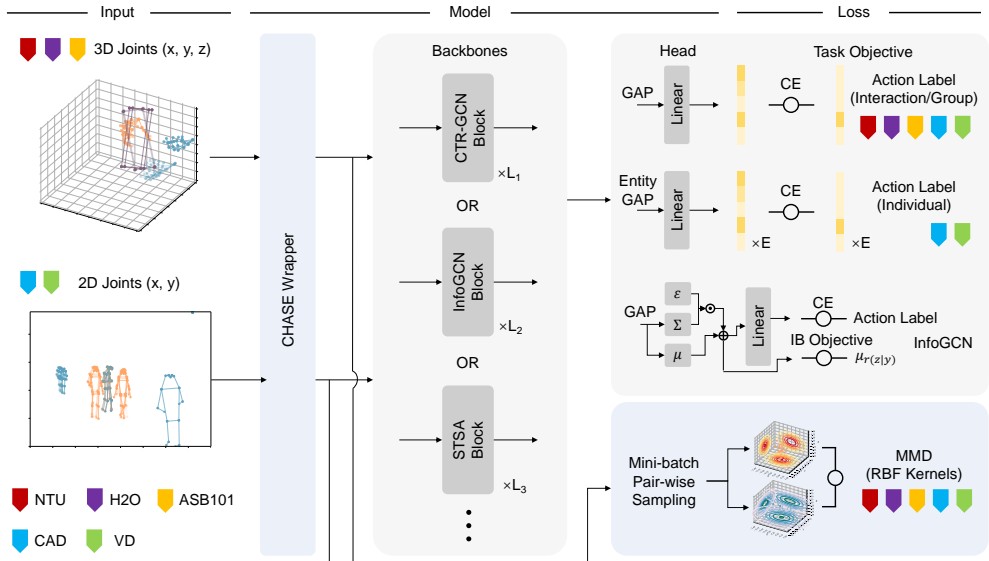

Figure 7: **Implementation details for different models and benchmarks.** CHASE can be adapted to various backbones including CTR-GCN [36], InfoGCN [37], STSA-Net [40], and HD-GCN [38]. We implement linear classification heads for all the models and benchmarks, except for InfoGCN [37]. Individual labels are utilized as an auxiliary classification objective to further improve the group action recognition performance in CAD [85] and ASB101 [12].

**Jensen-Shannon Divergence (JSD):**

$$JSD(P\|Q) = \frac{1}{2}D(P\|M) + \frac{1}{2}D(Q\|M),$$ (16)

where $M = \frac{1}{2}(P + Q)$ is a mixture distribution of $P$ and $Q$.

**Bhattacharyya Distance (BD):**

$$D_B(P, Q) = -\ln \sum_{x \in \mathcal{X}} \sqrt{P(x)Q(x)}.$$ (17)

**Hellinger Distance (HD):**

$$H(P, Q) = \frac{1}{\sqrt{2}} \sqrt{\sum_{x \in \mathcal{X}} (\sqrt{P(x)} - \sqrt{Q(x)})^2}.$$ (18)

**Maximum Mean Discrepancy (MMD):**

$$\text{MMD}(P, Q) = \sup_{\|f\|_{\mathcal{H}} \leq 1} \left(\mathbb{E}[f(X)] - \mathbb{E}[f(Y)]\right),$$ (19)

where $\mathbb{E}[f(\cdot)]$ is the expectation of any function $f$ in the RKHS $\mathcal{H}$.

When adopting these metrics in our experiments, the distributions are generated by the kernel density estimation (KDE). We sample the points 30 times with different seed initializations and report the averaged measurements.

# E    Implementation & Configuration Details

In this section, we provide more details of our experimental setup and model implementation for each benchmark. Experiments are conducted with 8 GeForce RTX 3070 GPUs (GPU Memory: 8GB), using torch version 1.9.0+cu111, torchvision version 0.10.0+cu111, and CUDA version 11.4. CTR-GCN [36], InfoGCN [37], STSA-Net [40] and HD-GCN [38] are chosen as our baseline models. To ensure fair comparisons, we adopt single intra-skeleton modality without multi-modality fusion, following [35]. Our implementation details are illustrated in Fig. 7.

### E.1 Dataset-related Configurations

**NTU Mutual 11** [41] **& NTU Mutual 26** [42]. X-Sub and X-View criteria [41] are adopted in NTU Mutual 11, while X-Sub and X-Set criteria [42] are used in NTU Mutual 26. We evaluate models with only 3D joint inputs, applying data augmentations such as random rotation and spatial shift. During training and testing, we employ temporal cropping and resizing, adjusting based on the number of valid frames. Notably, we use distinct percentage intervals for training (0.5,1) and testing (0.95). For experiments conducted on the entire NTU-RGB+D 120 dataset, we maintain identical settings as those used for NTU Mutual 26.

**H2O** [13]. We follow the training, validation, and test splits described in [13]. To maintain consistency in GCNs, we use the hand graph structure, originally designed for human hands, for both hand entities and object entities. Models are evaluated with only 3D joint inputs. The same augmentations as NTU Mutual 26 are adopted in this benchmark.

**ASB101** [12]. We follow the training, validation, and test splits outlined in [12] for evaluations. 1,380 Fine-grained actions (verb & noun) are adopted as labels in experiments. We evaluate models with only 3D joint inputs. The same augmentations as NTU Mutual 26 are adopted in this dataset, except that the training percentage interval of the temporal cropping and resizing is set to (0.75,1).

**CAD** [85]. We adopt the same data augmentations, group action categories, individual labels and train-test split in [95]. But different from [95], only 2D joint coordinates are used in our experiments. Individual labels are used as an auxiliary classification objective, as presented in Fig. 7.

**VD** [86]. We follow the same data augmentations, group action categories, individual labels and Original train-test split in [95]. But different from [95], only 2D joint coordinates are used as input features in experiments. Besides, individual labels are leveraged as an auxiliary classification objective to further improve the group action recognition performance, as shown in Fig. 7. The volleyball position is represented as $X \in \mathbb{R}^{T \times C}$. To ensure inter-entity consistency, we apply padding to fit the shape $X \in \mathbb{R}^{C \times T \times J \times 1}$. To maintain consistency in GCNs, we employ human body graph structure for both human body entities and volleyball entities.

### E.2 Model-related Configurations

**CHASE**. We set default segment size $(1, 1, 1)$, $C_1 = 64$, $C_2 = 8$, $M = 1$ and $\lambda = 0.1$ in CHASE. In all experiments, we maintain consistent configurations for baseline models to ensure fair comparisons between models incorporating CHASE and their respective vanilla counterparts. To avoid unnecessary verbosity, we present implementation specifics solely for NTU Mutual 26. For training details pertaining to other benchmarks, please refer to the code repository at `https://github.com/Necolizer/CHASE`.

**CTR-GCN** [36]. Cross entropy is used as the recognition loss function. SGD optimizer is used with Nesterov momentum of 0.9, a initial learning rate of 0.1 and a decay rate 0.1 at the 80th and 100th epoch. Batch size is set to 64. With the first 5 warm-up epochs, the training process is terminated after 110 epochs.

**InfoGCN** [37]. By taking $k = 1$, InfoGCN adopts their definition of 1-hop Bones [37] as input. Cross entropy is used as the loss function with label smoothing factor 0.1 and temperature factor 1.0. The information bottleneck objective [37] is also employed as the auxiliary loss. Following [37], we set $\lambda_1 = 0.1$, $\lambda_2 = 0.0001$, and the $\mu_r(z|y)$ of each action class as random orthogonal vectors with a scale of 3. Diverging from conventional backbones, we substitute the standard classification head with the InfoGCN head, depicted in Fig. 7. SGD optimizer is used with Nesterov momentum of 0.9, a weight decay of 0.0005, a initial learning rate of 0.1 and a decay rate 0.1 at the 90th and 100th epoch. Batch size is set to 120. With the first 5 warm-up epochs, the training process is terminated after 110 epochs.

**STSA-Net** [40]. Cross entropy is used as the loss function with label smoothing factor 0.1 and temperature factor 1.0. SGD optimizer is used with Nesterov momentum of 0.9, a initial learning rate of 0.1 and a decay rate 0.1 at the 60th and 90th epoch. Batch size is set to 64. With the first 5 warm-up epochs, the training process is terminated after 110 epochs.

**HD-GCN** [38]. We set $CoM = 1$ in the hierarchy graph generation and evaluate on this setting. Cross entropy is used as the loss function. SGD optimizer is used with Nesterov momentum of 0.9, a

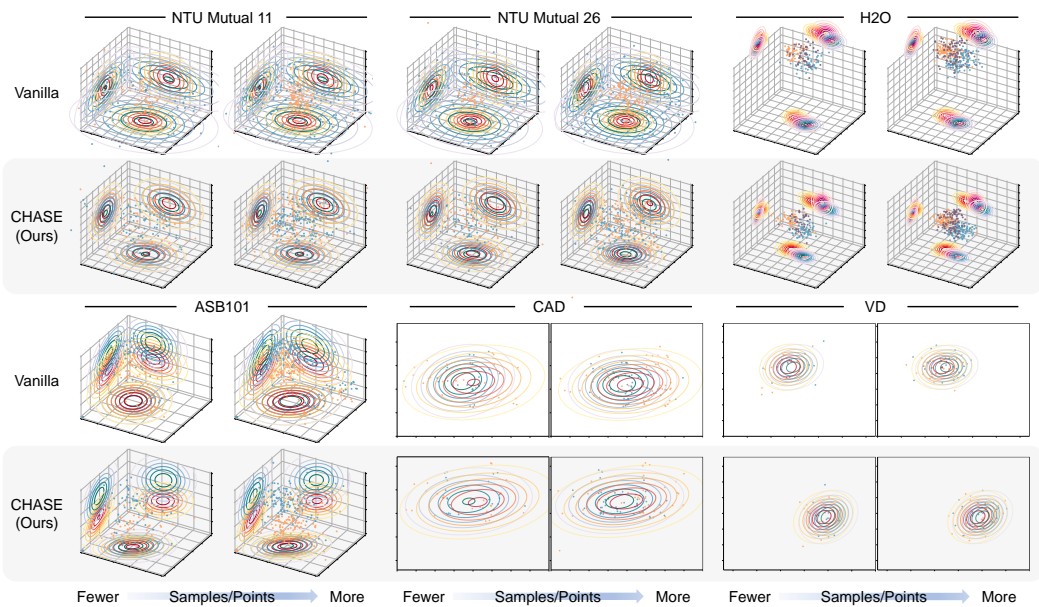

Figure 8: **Qualitative results on multi-entity action recognition datasets**. For visual clarity, we display 2 or 3 estimated entity distributions for each test set. Each subplot shows the projection of the multi-variant normal distributions generated by mean vectors and covariance matrices. Different entity distributions are denoted by distinct colors. CHASE effectively mitigates inter-entity distribution discrepancies while preserving potential entity orderliness in these datasets.

initial learning rate of 0.1 and a decay rate 0.1 at the 80th and 100th epoch. Batch size is set to 64. With the first 5 warm-up epochs, the training process is terminated after 110 epochs.

## F   Supplementary Experimental Results

**More Analysis on Table 1.** In Table 1, we observe that CHASE yields varying degrees of accuracy improvement across different baseline models and benchmarks. The performance gains are influenced by both the backbone models and the datasets, as CHASE functions as an additional normalization step that mitigates data bias introduced by inter-entity distribution discrepancies. For baseline backbones, this is owing to differences in their backbone architecture design, parameter count and training objective. For example, STSA-Net [40] is a relatively large backbone based on transformer architecture, which does not rely the prior definition of the skeleton graphs. Its adaptability to different graph structures makes it outweigh GCN-based backbones in H2O benchmark. Another example is InfoGCN [37], which leverages an auxiliary information bottleneck objective in its training. Though this method is proven more effective than the other backbones in some person-to-person interaction settings, it doesn't ensure better performance in hand-to-object interaction and group activity benchmarks. For different benchmarks, it is owing to differences in data scale and label space (see Table 7). For example, ASB101 is an extremely challenging benchmark for its over 80,000 samples and 1,380 target categories. Therefore the accuracy improvement is modest compared with the other benchmarks.

**More Qualitative Results.** Fig. 8 visualizes how CHASE works with multi-entity skeletal sequences. By integrating CHASE, different entity distributions become more similar in both aspects of mean and covariance. It demonstrates that CHASE can lower the inter-entity distribution discrepancy, especially obviously in NTU Mutual 11, NTU Mutual 26, CAD, whose entities have no orderliness. For H2O, ASB101 and VD, whose entities are characterized by an intrinsic order (e.g. left hands, right hands, left-side volleyball players, right-side volleyball players, and objects), CHASE can also preserve their orderliness by letting the distributions be similar but different.

**Evaluations on Mixed Recognition of Single-Entity & Multi-Entity Actions.** Table 5 concludes the action recognition results on the entire NTU RGB+D 120 [42]. By integrating CHASE, the baseline model gets accuracy improvement by 0.41% and 0.05% on X-Sub and X-Set, respectively.

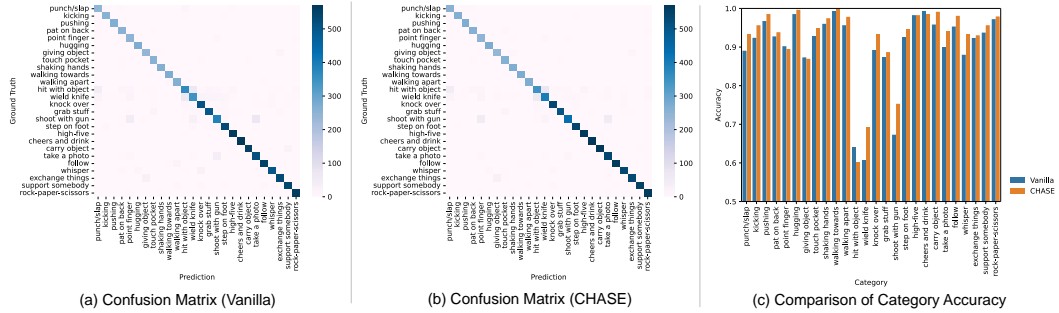

(a) Confusion Matrix (Vanilla)     (b) Confusion Matrix (CHASE)     (c) Comparison of Category Accuracy

Figure 9: **Confusion Matrices & Comparison of Category-level Accuracy on NTU Mutual 26 X-Sub**. (a) Confusion matrix of the vanilla CTR-GCN. (b) Confusion matrix of CTR-GCN with CHASE. (c) We present a detailed comparison of the category-level accuracy between the vanilla CTR-GCN (blue) and CTR-GCN with CHASE (orange). These results demonstrate the effectiveness of CHASE by improving recognition accuracy for most categories.

This suggests that CHASE is effective even in mixed recognition settings. However, the improvement is modest because single actions are the dominant category in this dataset.

**Analysis on Efficiency**. We analyse CHASE's efficiency on NTU Mutual 26 configurations. Our proposed CHASE is implemented as a backbone wrapper, adaptable to a variety of single-entity action models. As presented in Table 8, the number of trainable parameters is about 26.37 k, which only increases number of backbone's parameters by 1%-2%. We can approximate that the number of trainable parameters is increased by $(U + 1 + C_2) \times C_1 + C_2 \times U$. For computational complexity, the FLOPs of CHASE is approximately 2.50 M. This analysis proves that CHASE is both efficient and lightweight for benefiting multi-entity action learning.

Table 8: CHASE Trainable Parameters

| Method | # Param. (M) |
|---|---|
| CTR-GCN [36] | 1.44 |
| + CHASE | $1.46^{\uparrow 1.83\%}$ |
| InfoGCN [37] | 1.54 |
| + CHASE | $1.57^{\uparrow 1.96\%}$ |
| STSA-Net [40] | 4.13 |
| + CHASE | $4.16^{\uparrow 0.60\%}$ |
| HD-GCN [38] | 1.65 |
| + CHASE | $1.68^{\uparrow 1.60\%}$ |

**Segment Size of Squeeze Operator.** In CLB, which is also the mapping $\psi$ to weight matrix, the squeeze operator squeezes the tensor to a specific shape, denoted as $(T', J', E')$. This segment size determines the point set in a multi-entity action sequence to which ICHAS applies. For example, the segment size $(1, 1, 1)$ indicates that ICHAS applies to all the points, and $(1, 1, 2)$ means two different ICHAS apply to two entities separately. Table 9 evaluates various segment sizes of the squeeze operator, demonstrating that the global ICHAS with the segment size $(1, 1, 1)$ achieves the best performance compared with the other settings. Therefore, we choose the default segment size as $(1, 1, 1)$ in all the experiments. Besides, the reduction ratio between $C_1$ and $C_2$ is determined according to the experimental results in [98].

Table 9: Ablation on Segment Sizes

| $(T', J', E')$ | Acc (%) |
|---|---|
| **(1, 1, 1)** | $\mathbf{91.30}_{(\pm 0.22)}$ |
| (1, 1, 2) | $91.28_{(\pm 0.19)}$ |
| (2, 1, 1) | $91.20_{(\pm 0.04)}$ |
| (4, 1, 1) | $91.03_{(\pm 0.09)}$ |
| (1, 5, 1) | $91.23_{(\pm 0.05)}$ |

**Weight $\lambda$ for MPMMD.** Fig. 10 evaluates different values of the trade-off weight factor $\lambda$ in Eq. 12, varying from $10^{-2}$ to $10^0$. On NTU Mutual 26, it achieve the best performances when adopting $\lambda = 0.1$ for MPMMD loss. It corresponds to our claim that MPMMD is an auxiliary objective to guide discrepancy minimization, additional to the recognition task objective. We also conduct experiments with MPMMD across a variety of $M$ values but find insignificant differences in performance. Hence, we set $M = 1$ for computational efficiency.

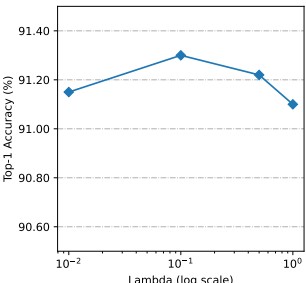

Figure 10: **Ablation study on different $\lambda$s for MPMMD.**

**Analysis on Confusion Matrix & Category-level Accuracy.** We present the confusion matrices of the vanilla CTR-GCN and CTR-GCN with CHASE in Fig. 9 (a) & (b). It indicates that our proposed CHASE is able to assist the backbone to differentiate similar multi-entity actions better. Fig. 9 (c) reports the category-level accuracy for 26 kinds of person-to-person interactions. We observe that in very few categories, their performance slightly drops. One possible

Table 10: Analysis of Performances with Test-Time Skeleton Noises and Masking

| Test-Time | + Noise (%) | | + Mask (%) | |
|---|---|---|---|---|
| | $\sigma = 10^{-3}$ | $\sigma = 10^{-2}$ | $p_m = 10^{-2}$ | $p_m = 10^{-1}$ |
| CTR-GCN [36] | $88.55_{(\pm 0.03)}$ | $80.72_{(\pm 0.03)}$ | $81.15_{(\pm 0.13)}$ | $56.37_{(\pm 0.13)}$ |
| **+ CHASE** | $\mathbf{91.24}_{(\pm 0.01)}$ | $\mathbf{82.53}_{(\pm 0.08)}$ | $\mathbf{88.57}_{(\pm 0.03)}$ | $\mathbf{60.65}_{(\pm 0.07)}$ |

reason is that these actions, like *point finger*, rely heavily on cues from the individual movements of one of the entities, instead of the multi-entity interactions. This might not be addressed by mitigating inter-entity distribution discrepancies. But it could still conclude from Fig. 9 (c) that adopting CHASE can achieve accuracy improvement for most of the categories, e.g. *wield knife* (+8.50%), *shoot with gun* (+8.00%), *whisper* (+5.39%), and *punch/slap* (+4.37%).

**Analysis of Performances with Test-Time Skeleton Noises and Masking.** To evaluate the robustness, we intentionally corrupt the multi-entity skeleton sequences with noise and masking during the inference phase. This aims at resembling possible skeleton occlusions or estimation errors during the test time. The noises $X_n \sim \mathcal{N}(\mu, \sigma^2)$ used in this experiment are normally distributed with mean $\mu = 0$ and standard deviations $\sigma = 10^{-3}, 10^{-2}$. Masking strategies are randomly masking the multi-entity skeleton sequences with probabilities $p_m = 10^{-2}, 10^{-1}$. Table 10 reports the averaged top-1 accuracy and its standard deviation in runs with several seed initializations for noises and masks. For recognizing multi-entity actions with corrupted test-time inputs, CTR-GCN integrating with CHASE consistently outperforms the vanilla counterpart, showcasing its robustness.

## G   Limitations & Broader Impacts

This work proposes the Convex Hull Adaptive Shift for Multi-Entity Action Recognition to resolve the inter-entity distribution discrepancies. Although CHASE offers a generic framework for various types of multi-entity actions, such as person-to-person interactions, hand-to-object interactions, hand-to-hand interactions and group activities, its application to single-entity actions warrants further investigation. One potential approach is to consider different parts as multiple entities, such as treating different limbs of a human body as distinct entities. Moreover, it's promising to apply CHASE-like designs to the recently-developed human-centric foundation models [17, 29, 30, 31, 32, 33, 34] to enhance their performances on multi-entity skeletal data. These areas of exploration are left for future research.

This paper focuses on multi-entity action recognition, a field with broad applications in physical human-robot interaction, social scene understanding, multi-agent systems, surveillance, healthcare monitoring, etc [24, 22, 25, 26, 27, 28, 23]. Our work contributes to advancements in these domains by enhancing efficient and effective skeleton-based learning of multi-entity actions. Although the use of human-centered data can pose privacy concerns, our study utilizes only skeletal data estimated from sensors or RGB videos, thus mitigating potential privacy and ethical issues.

## H   Licenses for Used Assets

Datasets:

- NTU Mutual 11 / NTU RGB+D [41]): Custom (research-only, non-commercial, attribution) [2]
- NTU Mutual 26 / NTU RGB+D 120 [42]: Custom (research-only) [3]
- H2O [13]: Custom (research-only, non-commercial) [4]
- Assembly101 [12]: Creative Commons Attribution-NonCommercial 4.0 International License [5]
- Collective Activity Dataset [85]: Unknown

---

[2] http://rose1.ntu.edu.sg/Datasets/requesterAdd.asp?DS=3
[3] http://rose1.ntu.edu.sg/Datasets/actionRecognition.asp
[4] https://h2odataset.ethz.ch/
[5] http://creativecommons.org/licenses/by-nc/4.0/

- Volleyball Dataset [86]: BSD 2-Clause license [6]

Models:

- CTR-GCN [36]: Creative Commons Attribution-NonCommercial 4.0 International License [7]
- InfoGCN [37]: Unknown
- STSANet [40]: MIT License [8]
- HD-GCN [38]: MIT License [9]

---

[6]https://github.com/mostafa-saad/deep-activity-rec/blob/master/LICENSE

[7]https://github.com/Uason-Chen/CTR-GCN/blob/main/LICENSE

[8]https://github.com/heleiqiu/STTFormer/blob/main/LICENSE

[9]https://github.com/Jho-Yonsei/HD-GCN/blob/main/LICENSE

