# OpenReview forum: "CHASE: Learning Convex Hull Adaptive Shift for Skeleton-based Multi-Entity Action Recognition"
_NeurIPS.cc/2024/Conference — NeurIPS 2024 poster_

### Official Review · Reviewer_WotV · 2024-07-09

**Soundness:** 3
**Presentation:** 3
**Contribution:** 3
**Rating:** 6
**Confidence:** 4

**Summary:**

This paper tackled the issue of inter-entity distribution discrepancies in multi-entity action recognition. The authors proposed convex hull adaptive shift method to minimize the cross entity discrepancies, where CLB and MPMMD are proposed to assist the learning procedure. The method is verified to be effective among various datasets and backbones.

**Strengths:**

1.This paper proposed an interesting idea by using implicit convex hull as constraints to achieve adaptive coordinate shift.

2.The proposed approach is verified to be effective on various datasets and backbones.

3.This method can serve as a good contribution to the skeleton-based human action recognition community.

**Weaknesses:**

1. The introduction section should  be improved. For example on line 54, why do we need to achieve the discrepancy minimization? on line 52, why do we need to achieve sample adaptive coefficients? The motivation should be highlighted. The links among these proposed items should be also improved on line 57-60 and need more insights.


2. The novelty of the CLB is limited. The format of attributes learning shown in Eq.8 is commonly used to construct concept learners. What is the difference between the concept bottleneck [1] and the CLB? If you use the concept learner from some existing works, e.g., [2], will it be better than CLB?


[1] Shin S, Jo Y, Ahn S, et al. A closer look at the intervention procedure of concept bottleneck models[C]//International Conference on Machine Learning. PMLR, 2023: 31504-31520.


[2] Wang B, Li L, Nakashima Y, et al. Learning bottleneck concepts in image classification[C]//Proceedings of the ieee/cvf conference on computer vision and pattern recognition. 2023: 10962-10971.

3. More insights should be given in Section 4.2. Why does the proposed method help? The authors are encouraged to enrich the analysis.
The authors are encouraged to discuss the computational complexity brought by the proposed method.

4. The authors are encouraged to discuss the computational complexity brought by the proposed method.

**Questions:**

1. Improving the Introduction Section:

    a. Why is it necessary to achieve discrepancy minimization (line 54)?

    b.Why do we need to achieve sample adaptive coefficients (line 52)?

    c. Can the authors highlight the motivation behind these needs and improve the links among the proposed items (lines 57-60) with more insights?


2. Novelty of the CLB:

    a. How does the concept bottleneck (CLB) differ from the commonly used format of attributes learning in concept learners, such as in Eq. 8?

    b. What are the differences between the CLB and the concept bottleneck models discussed in Shin et al. (2023)?

    c. If the concept learner from existing works (e.g., Wang et al. (2023)) were used, would it perform better than the CLB?

3. Insights in Section 4.2:

    a. Why does the proposed method provide benefits?

    b. Can the authors provide a more detailed analysis to enrich Section 4.2?

4. Computational Complexity:

    a. What is the computational complexity introduced by the proposed method?

**Limitations:**

yes in supplementary

---

> ### Author Rebuttal · Authors · 2024-08-06
>
> Thanks for your encouraging and constructive comments. We appreciate your recognition of the methodology, experimental results, and contributions. We understand that your concerns may arise from the need for greater clarity and detail in presenting our motivations and methodology. Below, we address these issues one by one and have clarified them in the revised paper. We hope our response will address your concerns effectively.
>
> ----
> **Q1a.** *Why is it necessary to achieve discrepancy minimization?*
>
> These discrepancies can introduce bias into the backbone models, leading to suboptimal optimization and performance, as shown in the upper histogram of Fig. 1(b). The lower histogram of Fig. 1(b) indicates that minimizing these discrepancies helps reduce bias in the classification backbone, thereby enhancing its recognition performance on this task.
>
> ----
> **Q1b.** *Why do we need to achieve sample adaptive coefficients?*
>
> Achieving sample adaptive coefficients is a necessary step to adaptively reposition each skeleton sequence. As shown in Table 4 in our main draft, the recognition accuracy drops to 91.20\% **without sample-adaptive coefficients**. It implies that sample-adaptive weight representations are important to achieve better recognition performance.
>
> ----
> **Q1c.** *Can the authors highlight the motivation behind these needs and improve the links among the proposed items with more insights?*
>
> We have revised the paragraph to clearly present our motivation and the connections among the proposed components:
>
> "... **Specifically, CHASE consists of a learnable parameterized network and an auxiliary objective.** The parameterized network can achieve **plausible and sample-adaptive** repositioning of skeleton sequences through two crucial components. First, the Implicit Convex Hull Constrained Adaptive Shift (ICHAS) ensures that the new origin of the coordinate system is within the skeleton convex hull. Second, the Coefficient Learning Block (CLB) provides a lightweight parameterization of the mapping from skeleton sequences to their specific coefficients in ICHAS. Moreover, **to guide the optimization of this network for discrepancy minimization**, we propose the Mini-batch Pair-wise Maximum Mean Discrepancy (MPMMD) as the additional objective. This loss function quantifies ... **In conclusion, CHASE works as a sample-adaptive normalization method to mitigate inter-entity distribution discrepancies, which can reduce bias in the subsequent backbone and enhance its multi-entity action recognition performance.**"
>
> ----
> **Q2a \& b.** *What is the difference between the concept bottleneck (Shin et al. (2023)) and the CLB?*
>
> After carefully reviewing the relevant literature [1,2], we have identified several key differences between our CLB and concept bottleneck models (CBMs):
>
> 1. **Motivation**: CBMs are primarily designed to enhance interpretability [1]. Different from CBMs, our CLB is the parameterization of a mapping, which maps the input skeleton sequence to the coefficient matrix. Thus, **while CBMs focus on interpretability, CLB is geared towards adaptively adjusting the skeleton sequence representation to reduce inter-entity discrepancies**.
> 2. **Inputs**: CBMs typically require input data $x\in \mathbb{R}^{d}$, binary concepts $c\in \{0,1\}^{k}$, and target responses $y\in Y$ [1,2]. However, **in multi-entity action recognition tasks, there are no binary concepts to work with**, making the direct application of CBMs unsuitable for our task.
> 3. **Architecture Design**: The architecture of CBMs, as implemented in [1], is expressed as $\hat{y} = f(\delta(g(x))$, where $\delta$ denotes ReLU, and $f,g$ denotes inceptionv3 \& MLP. In [2], it's based on slot attention and is formulated as $a_k = \phi(Q(c_k)^TK(F'))$, where $Q,K$ are nonlinear transformations. Our CLB, however, uses a different architecture: $W=\psi(X)=W_3\delta(W_2\phi(W_1X+b))$. This distinction highlights that **our CLB does not rely on inceptionv3+ReLu+MLP or slot attention mechanisms**.
>
> ----
> **Q2c.** *If the concept learner from existing works (e.g., Wang et al. (2023)) were used, would it perform better than the CLB?*
>
> CBMs require binary concepts [1,2], which is **not applicable to the multi-entity action recognition task**. Therefore, adapting a concept learner from existing works to our task is not feasible. Given this, it is not possible to directly compare their performance with our CLB.
>
> ----
> **Q3a.** *Why does the proposed method provide benefits?*
>
> By adaptively shifting skeleton sequences, CHASE effectively mitigates inter-entity distribution discrepancies in multi-entity skeletal data, which can unbias the subsequent classification backbone and boost their recognition performance.
>
> ----
> **Q3b.** *Can the authors provide a more detailed analysis to enrich Section 4.2?*
>
> Yes, we have provided a more detailed analysis of Section 4.2, as mentioned in Section F (line 746-757) of our initial submission:
>
> "In Table 1, we observe that CHASE yields varying degrees of accuracy improvement across different baseline models and benchmarks. The performance gains are influenced by both the backbone models and the datasets, as CHASE functions as an additional normalization step that mitigates bias in the backbone introduced by inter-entity distribution discrepancies.
> For baseline backbones, this is owing to differences in their backbone architecture design, parameter count and training objective. For example, ... "
>
> ----
> **Q4:** *What is the computational complexity introduced by the proposed method?*
>
> We have discussed the computational complexity in Section 4.3 (line 270-275) \& Section F (line 774-785) of our initial submission. As presented in Table 9, the number of trainable parameters is about **26.37 k**. We can approximate that the number of trainable parameters is increased by $(U+1+C_2)\times C_1 + C_2 \times U$. For computational complexity, the FLOPs of CHASE is approximately **2.50 M**.

---

> ### Author Response · Authors · 2024-08-12
> **Looking Forward to Further Discussions**
>
> Dear Reviewer,
>
> Thanks again for your insightful comments on our paper.
>
> We have submitted the response to your comments and the global response with a PDF file. Please let us know if you have additional questions so that we can address them during the discussion period. We hope that you can consider rasing the score.
>
> Thank you

---

> > ### Comment · Reviewer_WotV · 2024-08-12
> > **To the authors**
> >
> > Dear authors,
> >
> > thank you very much for your rebuttal. I think most of my concerns are handled and I will improve my score to 6.
> >
> > Best,

---

> > > ### Author Response · Authors · 2024-08-13
> > > **Thank You for Your Positive Feedback and Consideration**
> > >
> > > Dear Reviewer,
> > >
> > > Thank you for your positive feedback and for taking the time to review our rebuttal. We're glad that our responses addressed your concerns, and we appreciate your willingness to improve the score. Thank you again for your thoughtful consideration.
> > >
> > > Best regards,

---

### Official Review · Reviewer_16iH · 2024-07-10

**Soundness:** 2
**Presentation:** 3
**Contribution:** 2
**Rating:** 6
**Confidence:** 4

**Summary:**

This paper proposes CHASE, a multi-entity skeleton data augmentation/preprocessing technique, to mitigate inter-entity distribution gaps and improve the multi-entity action recognition. Specifically, the authors formulate a new constraint called ICHAS, design a lightweight block CLB to learn the nonlinear mapping from input to the weight matrix in ICHAS, and introduce an objective to guide the discrepancy minimization in CLB training. The authors conduct comprehensive experiments to show the effectiveness of the method.

**Strengths:**

a) The paper is well-written, easy to understand, and well-organized.

b) The method is well-motivated, well-ablated, and the experiments are presented clearly.

**Weaknesses:**

a)	There’re some typos need to be revised carefully, e.g. l131 “be be”, Table 4 “-68.56”.

b)	Regarding clarity, the captions of Figures and Tables can be improved.

**Questions:**

a)	Figure 2 is very informative and good for understanding the whole method. However, there are still some notations needed to be explained, e.g. the green and red circles under the two skeleton coordinates. It would be helpful if the authors introduce the method details (in sections 3.1 to 3.3) with references to specific parts of Figure 2.

b)	It’s good to have multiple runs and report the standard deviation. How many seed initializations exactly do the authors use?

c)	From Table 1, the performance margin varies a lot. On H2O and CAD, the top-1 accuracies have increased by over 9%. Could the authors explain or provide some discussion on this point?

d)	From Table 7 in the appendix, it seems like the authors use 25 ground-truth 3D joints as skeleton inputs for NTU-60 and NTU-120 datasets. Since 2D estimated joints tend to achieve better action recognition performance in dominant models, do the authors verify CHASE using 17 estimated 2D joints with COCO layout as input?

e)	Table 6 shows the mixed recognition results on the entire NTU-120 dataset with X-Sub setting. Table 8 in the appendix shows the mixed recognition results on NTU-120 X-Sub and X-Set settings. Why not replace Table 6 with Table 8? Similar to Table 5 and Table 9, why not replace Table 5 with Table 9 or integrate the number of parameters into Table 1 (may discard the column of ‘Venue’)?

Small details (no need to address them, just suggestions):

a)	Regarding equations (9) and (10), it would be better to clarify the definitions of sup(*) and C(E,2).

**Limitations:**

Although the method has been verified on six benchmarks, these datasets are relatively small, with the number of categories ranging from 4 to 36, except Assembly 101 has 1380 action categories. However, the performance gain on Assembly 101 is very small (<=0.21%). The reviewer has some concerns about the generalization of this method.

---

> ### Author Rebuttal · Authors · 2024-08-06
>
> Thanks for your detailed and constructive comments. We appreciate your recognition of the writing, motivation and experiments. We understand that some of your concerns may stem from clarity and experimental details. Below, we address these issues one by one and have clarified them in the revised paper. We hope our response will address your concerns.
>
> ---
> **Q1:** *There’re some typos need to be revised carefully, e.g. l131 “be be”, Table 4 “-68.56”. Regarding clarity, the captions of Figures and Tables can be improved.*
>
> We are grateful for your indications of typos and suggestions for improving clarity. We have checked and revised the manuscript for these issues. For example, we have revised the caption of Figure 1 to provide a more concise illustration of our motivation.
>
> ---
> **Q2:** *There are still some notations needed to be explained in Figure 2. It would be helpful if the authors introduce the method details (in sections 3.1 to 3.3) with references to specific parts of Figure 2.*
>
> We have added notation explanations to Figure 2 in the revised version. For example, the green and red circles represent feasible and infeasible $\vec{p^*}$ as formulated by Eq. 3, respectively. Additionally, we have provided details in sections 3.1 to 3.3 with references to Figure 2 to facilitate understanding.
>
> ---
> **Q3:** *How many seed initializations exactly do the authors use?*
>
> We adopt three seed initializations in most settings, which means three runs from training to evaluation.
>
> ---
> **Q4:** *Why does the performance margin vary a lot?*
>
> **As CHASE functions as an additional normalization step, the performance gains are influenced by the bias both in datasets and introduced in backbone models, especially their different architecture design, data scale, and label space.** We have discussed it in detailed in Section F (line 746-757) of our submission:
>
> "In Table 1, we observe that CHASE yields varying degrees of accuracy improvement across different baseline models and benchmarks. ... For different benchmarks, the variations in performance margins are due to differences in data scale and label space (see Table 7). For example, ... In contrast, H2O and CAD are relatively small in data scale. The significant accuracy increase implies that our CHASE can unbias the subsequent backbone more effectively with limited training data better."
>
> ---
> **Q5:** *Do the authors verify CHASE using 17 estimated 2D joints with COCO layout as input?*
>
> We have evaluated CHASE using 17 estimated 2D joints as input on the **CAD and VD benchmarks**, as visualized in Figure 3 of our initial submission. Additionally, we have conducted experiments on the **NTU Mutual 26 dataset** using 17 estimated 2D joints. The results are reported in the following table. It demonstrates that our proposed CHASE also enhances the performance of the vanilla counterparts when using 2D joint inputs.
>
> Method | X-Sub (\%) | X-Set (\%)
> :----|:----:|:----:
> CTR-GCN | ${90.10}_{(\pm0.05)}$ | ${91.35}_{(\pm0.15)}$
> **+ CHASE (Ours)** | **${90.56}_{(\pm0.09)}$** | **${92.38}_{(\pm0.81)}$**
>
> ---
> **Q6:** *Why not replace Table 6 with Table 8? Similar to Table 5 and Table 9, why not replace Table 5 with Table 9 or integrate the number of parameters into Table 1?*
>
> Due to the page limit of the initial submission, we included excerpted versions (Table 5 \& 6) in the main draft and placed the full versions (Table 8 \& 9) in the supplementary material. We will replace the excerpted versions with the full tables, if the paper is accepted, as additional content pages will be allowed. Integrating the number of parameters into Table 1 might cause confusion, as the number of model parameters varies across different benchmarks. Therefore, we opted to keep the parameter details separate to maintain clarity.
>
> ---
> **Q7:** *It would be better to clarify the definitions of sup(\*) and C(E,2).*
>
> We have clarified the definitions in the revised version. The notation $\sup(\cdot)$ stands for the supremum. In Eq. (9), $\sup(\cdot)$ is used to denote the maximum value that the expression $\mathbb{E}[f(x)]-\mathbb {E} [f(y)]$ can attain, where the function $f$ is taken from a specific class of functions. $C(E,2)$ stands for the combination of $E$ things taken 2 at a time without repetition. In Eq. (10), $C(E,2)$ is used to denote the total count of possible entity pairs.
>
> ---
> **Q8:** *These datasets are relatively small ... The reviewer has some concerns about the generalization of this method.*
>
> We provide the following reasons to support the generalization capability of CHASE:
>
> 1. We have compared our approach with recent methods in multi-entity action recognition, as shown in the table below. **These methods are typically evaluated on $\leq 5$ datasets, with some using only a subset of the datasets we have utilized.** In contrast, we have verified CHASE across six diverse benchmarks, including person-to-person interactions, hand-to-object interactions, and group activities. Our approach outperforms these methods, as reported in Table 1 of our submission.
>
> 2. Additionally, we have evaluated CHASE on ASB101, which is the largest and most challenging benchmark for this task. **Given the complexity of this dataset, achieving substantial accuracy improvements is inherently difficult.** Although the improvement on ASB101 is modest, it still demonstrates the effectiveness of CHASE in a challenging scenario.
>
> Method | Venue | \#Dataset | Max \#Category | Datasets for Multi-entity Action Recognition
> :----|:----:|:----:|:----:|:----:
> IGFormer | ECCV'22 | 3 | 26 | NTU Mutual 11/26, SBU (200 samples)
> ISTA-Net | IROS'23 | 4 | 1380 | NTU Mutual 26, SBU (200 samples), ASB101, H2O
> H2OTR | CVPR'23 | 2 | 45 | H2O, FPHA (1175 samples)
> me-GCN | arXiv'24 | 3 | 1380 | NTU Mutual 11/26, ASB101
> EffHandEgoNet | arXiv'24 | 2 | 45 | H2O, FPHA (1175 samples)
> AHNet-Large | PR'24 | 5 | 26 | NTU Mutual 11/26, CAD, VD, PKU-MMD mutual
> **Ours** | - | 6 | 1380 | NTU Mutual 11/26, H2O, ASB101, CAD, VD

---

> ### Author Response · Authors · 2024-08-12
> **Looking Forward to Further Discussions**
>
> Dear Reviewer,
>
> Thanks again for your insightful comments on our paper.
>
> We have submitted the response to your comments and the global response with a PDF file. Please let us know if you have additional questions so that we can address them during the discussion period. We hope that you can consider rasing the score.
>
> Thank you

---

### Official Review · Reviewer_hFtj · 2024-07-12

**Soundness:** 2
**Presentation:** 2
**Contribution:** 2
**Rating:** 4
**Confidence:** 4

**Summary:**

This paper focuses on the interesting problem of the normalization strategy for multi-entity skeletons in skeleton-based action recognition. The proposed method is intuitive, and the authors provided detailed implementation details for reproduction. However, this work is unclear, and the experiments are unconvincing.

**Strengths:**

This paper focuses on the interesting problem of the normalization strategy for multi-entity skeletons in skeleton-based action recognition. The proposed method is intuitive, and the authors provided detailed implementation details for reproduction.

**Weaknesses:**

(1)	The purpose of the multi-entity action recognition task is not clear. Is it to recognize each individual’s action or to classify group activities? If the purpose varies across different datasets, please clarify this in the experiment. Additionally, I am curious whether the optimal normalization strategy differs for these two purposes. For example, the method used in S2CoM seems more suitable for recognizing each individual’s action.
(2)	What is the main difference between the proposed method and the simple strategy of shifting the origin of multi-entities to their common center? Please add a comparison experiment with this method.
(3)	Although the normalization strategy is an important trick and can bring significant improvement in the action classification task, I still have a concern about whether it is worth conducting an additional network to achieve this simple trick by introducing extra learnable parameters. Some heuristic strategies may be more efficient and general. Accordingly, can the proposed module be transferred among different datasets without retraining the module?

**Questions:**

What is the main difference between the proposed method and the simple strategy of shifting the origin of multi-entities to their common center? Please add a comparison experiment with this method.

**Limitations:**

(1)	The purpose of the multi-entity action recognition task is not clear. Is it to recognize each individual’s action or to classify group activities? If the purpose varies across different datasets, please clarify this in the experiment. Additionally, I am curious whether the optimal normalization strategy differs for these two purposes. For example, the method used in S2CoM seems more suitable for recognizing each individual’s action.
(2)	What is the main difference between the proposed method and the simple strategy of shifting the origin of multi-entities to their common center? Please add a comparison experiment with this method.
(3)	Although the normalization strategy is an important trick and can bring significant improvement in the action classification task, I still have a concern about whether it is worth conducting an additional network to achieve this simple trick by introducing extra learnable parameters. Some heuristic strategies may be more efficient and general. Accordingly, can the proposed module be transferred among different datasets without retraining the module?

---

> ### Author Rebuttal · Authors · 2024-08-06
>
> Thanks for your insightful comments. We appreciate your recognition of the motivation behind CHASE and the extensive implementation details for reproducibility. We understand that your concerns may stem from some misunderstandings and the presentation of ablation studies. Below, we address these issues one by one and have clarified them in the revised paper. We hope our response will address your concerns.
>
> ---
> **Q1:** *The purpose of the multi-entity action recognition task is not clear. Is it to recognize each individual’s action or to classify group activities? If the purpose varies across different datasets, please clarify this in the experiment. Additionally, I am curious whether the optimal normalization strategy differs for these two purposes.*
>
> We apologize for any confusion.
>
> 1. The purpose of the multi-entity action recognition task is consistently to **recognize the comprehensive actions performed by multiple entities**, rather than individual actions. Multiple entities can include human bodies, hands, and objects. This is discussed in Section 1 (lines 21-23) and Section A.1 (lines 613-619) of our submission. This goal aligns with many related works focusing on interactive actions and group activities [11,25,85]. To avoid misunderstandings, we have clarified this better in Section.1 of the revised version.
> 2. The aim **does not** vary across different datasets. All experiments are conducted with the purpose of recognizing multi-entity actions. Notably, in CAD and VD benchmarks, individual labels are leveraged as an auxiliary objective to further improve the group action recognition performance, as mention in line 708-709 and 712-713. This strategy is a common practice adopted by most models training on these two datasets [83,85].
> 3. Exploring whether the optimal normalization strategy differs for these two purposes is indeed an interesting topic. However, it is not the focus of this paper. We will delve into this issue in future work.
>
> ---
> **Q2:** *What is the main difference between the proposed method and the simple strategy of shifting the origin of multi-entities to their common center? Please add a comparison experiment with this method.*
>
> In our initial submission, we have reported the experimental result of the comparison of CHASE and this simple strategy in Table 2, highlighting the superior performance of CHASE.
>
> 1. **The main difference is that the common center of mass (CoM) of all entities is just a subset of the search space for** $\vec{p^*}$ **in CHASE**. As indicated in line 149-150 in our manuscript, their common CoM $\bar{\vec{p}}$ is in the open convex hull of $X$, proven by simply taking all $\tilde{\alpha}_i=1/U(1\leq i\leq U)$. Therefore, **shifting the origin of multi-entities to their common center is just one possible result among all possible ones in CHASE**.
> 2. While it is intuitive to shift the origin to their common center, **experimental results shows that this is not always the optimal choice**. In our main draft, we have reported the experimental result of the comparison you mentioned in Table 2 (the last two rows). Our proposed CHASE achieves 91.30\% top-1 accuracy on NTU 26 X-Sub benchmark, while this simple (denoted as S2CoM†) obtains 90.79\%. This comparison implies that adopting adaptive $\vec{p^*}$ for each skeleton sequence sample is superior to this intuitive approach.
>
> ---
> **Q3:** *I still have a concern about whether it is worth conducting an additional network to achieve this simple trick by introducing extra learnable parameters. Some heuristic strategies may be more efficient and general.*
>
> **Our proposed CHASE outperforms other alternative strategies**, as mentioned in Tables 2 of our submission. We acknowledge that there is an inevitable trade-off between efficiency and performance. However, we believe our proposed CHASE strikes a good balance in this trade-off. We have addressed your concerns with extensive ablation studies in Tables 2 and 9. In our initial submission, Table 2 presents a comparison with many alternative strategies, including BatchNorm, S2CoM, and data augmentations. Our CHASE achieves **the best recognition performance**, highlighting the advantage of adaptively shifting skeletons to unbias the subsequent classification backbone. Moreover, as shown in Table 9, the number of trainable parameters is approximately 26.37k, which **only increases the number of parameters by 1\%-2\%**. These findings show the advantages of adopting CHASE for this task.
>
> ---
> **Q4:** *Accordingly, can the proposed module be transferred among different datasets without retraining the module?*
>
> In most cases, **we can't directly transfer it without retraining, as the input shapes vary across different skeleton datasets**, indicated by Table 7. However, your insightful suggestion prompted us to investigate whether CHASE can be transferred if we align the input shapes of two datasets. We conducted an experiment on a modified version of the H2O dataset, aligning its input skeleton shape to the ASB101 dataset by discarding object poses and sampling 70 frames (as used in ASB101). The results in the following table demonstrate that both the frozen CHASE module (pretrained on ASB101) and the retrained CHASE module improve the performance of the CTR-GCN backbone. This implies the transferability of our proposed CHASE, provided the skeleton sequences are aligned.
>
> Method | Acc (\%) | $\Delta$ (\%)
> :-------- | :-----: | :-----:
> CTR-GCN | ${48.48}_{(\pm2.91)}$ | -
> \+ CHASE (retrained) | ${56.47}_{(\pm1.59)}$ | $+7.99$
> \+ CHASE (frozen, pretrained on ASB101) | ${56.61}_{(\pm4.41)}$ | $+8.13$

---

> > ### Comment · Reviewer_hFtj · 2024-08-13
> > **Rebuttal Comment**
> >
> > I keep my initial score, where the authors did not struggle to handle my concerns, like "transferred among different datasets without retraining the module". Besides, the multi-entity action recognition task is still confusing.

---

> ### Author Response · Authors · 2024-08-12
> **Looking Forward to Further Discussions**
>
> Dear Reviewer,
>
> Thanks again for your insightful comments on our paper.
>
> We have submitted the response to your comments and the global response with a PDF file. Please let us know if you have additional questions so that we can address them during the discussion period. We hope that you can consider rasing the score.
>
> Thank you

---

> ### Author Response · Authors · 2024-08-13
> **Clarifications on Transferability and Multi-Entity Action Recognition**
>
> Dear Reviewer,
>
> Thank you for your feedback. We apologize for not fully addressing your concerns in our initial response.
>
> 1. We have followed your suggestions, conducting the experiments to evaluate the transferability of our CHASE module across different datasets without retraining. Specifically, we first trained the CHASE + CTR-GCN backbone on the challenging ASB101 dataset, achieving a top-1 accuracy of 28.03%. **We then transferred the CHASE module to the H2O (two-hand version) dataset without retraining, where it achieved 56.61% accuracy.** **This result outperforms both retraining the module on H2O (56.47%) and training only the CTR-GCN backbone on H2O (48.48%).** These findings demonstrate that our proposed module can indeed be transferred among different datasets without retraining.
> 2. Multi-entity action recognition **aims to classify interactions involving multiple entities, which could be people, objects, or other elements within a scene.** Examples of such actions include *cheers and drink*, *exchanging things*, *walking apart,* and *talking*. Group activities are a subset of multi-entity actions [73, 75, 76]. Unlike traditional action recognition, which typically focuses on a single subject performing a single action, multi-entity action recognition addresses the complexity of interpreting actions that involve multiple participants or objects interacting simultaneously. We hope this clarification addresses the confusion.
>
> We appreciate your insights and hope this additional information helps to resolve your concerns.
>
> Best regards,

---

### Official Review · Reviewer_N2tt · 2024-07-12

**Soundness:** 3
**Presentation:** 2
**Contribution:** 3
**Rating:** 6
**Confidence:** 3

**Summary:**

The paper proposes a normalization method for skeleton-based multi-entity recognition based on finding the center of mass within the convex hull of the spatio-temporal domain of the point cloud defined by the skeletons over a sequence. The main idea is to "center" the world of skeletons to unbias the subsequent detector and boost their performance. Building on this motivation, the authors find that a fixed, learnable parameterized network can be used for that purpose, facilitating the inference. The experiments demonstrate that their proposed algorithm boosts performance over the corresponding baselines

**Strengths:**

The paper is technically sound and the authors follow a proper mathematical derivation that leads to the design of CHASE in a clever manner.

The paper includes an extensive supplementary material with code and further analysis that make the paper rather complete.

The method is elegant and simple, providing with a very efficient, lightweight network that shows provable performance on a broad variety of datasets. Ablation studies are conducted to validate the proposed parts, as well as to compare against other normalization alternatives.

The paper is well documented with an extensive coverage of related literature.

**Weaknesses:**

Overall I believe that the presentation should be improved, clearly stating the contribution and motivation for it. It takes a good read to understand that what the authors are proposing as a lightweight network is the result of mathematically deriving an iterative approach for normalization of skeletons within their convex hull. It should be clearly stated that the method aims to accompany other methods for multi-entity activity recognition by adding an extra normalization step, which consists of what is presented in Section 3.

Similarly, the sketch depicted in Figure 1 is a bit confusing and does not really illustrate what the authors aim to solve. I would suggest the authors to provide a clear motivation example that leads to their method. The caption in Fig. 1 is rather poorly written (I did not understand it at least).

**Questions:**

I would insist on the authors to please elaborate a bit better on the motivation and an example where former normalization leads to the classifiers to produce wrong results, with their method mitigating such problem.

---

> ### Author Rebuttal · Authors · 2024-08-06
>
> We appreciate your recognition of the methodology of CHASE, proper mathematical derivation, and extensive experiments. We understand that your concerns may stem from clarity and presentation of our contributions and motivation. Below, we address these issues one by one and have clarified them in the revised paper. We hope our response effectively addresses your concerns.
>
> ---
> **Q1:** *The presentation should be improved, clearly stating the contribution and motivation for it. It should be clearly stated that the method aims to accompany other methods for multi-entity activity recognition by adding an extra normalization step, which consists of what is presented in Section 3.*
>
> We deeply appreciate your time and thorough review of our paper. We apologize for any confusion in the presentation of our contribution and motivation. We have revised the final paragraph of the Introduction section to better articulate our contributions and motivation.
>
> The contributions of this paper are three-fold:
>
> 1. To the best of our knowledge, we are the first to investigate the issue of inter-entity distribution discrepancies in multi-entity action recognition. Our proposed method, Convex Hull Adaptive Shift for Multi-Entity Actions, effectively addresses this challenge. **Our main idea is to adaptively repositioning skeleton sequences to mitigate inter-entity distribution gaps, thereby unbiasing the subsequent backbones and boosting their performance.**
> 2. **Serving as an additional normalization step for backbone models, CHASE consists of a learnable network and an auxiliary objective.** Specifically, the network is formulated by the Implicit Convex Hull Constrained Adaptive Shift, together with the parameterization of a lightweight Coefficient Learning Block, which learns sample-adaptive origin shifts within skeleton convex hull. Additionally, the Mini-batch Pair-wise Maximum Mean Discrepancy objective is proposed to guide the discrepancy minimization.
> 3. Experiments on NTU Mutual 11, NTU Mutual 26, H2O, Assembly101, Collective Activity Dataset and Volleyball Dataset consistently verify our proposed method by enhancing performance of single-entity backbones in multi-entity action recognition task.
>
> Our motivation is:
>
> When using *Vanilla* (a common practice), the estimated distributions of joints from different entities show significant discrepancies, as shown in Figure 1(a). These discrepancies can introduce bias into backbone models, leading to suboptimal optimization and poor recognition performance, as depicted in the upper histogram of Figure 1(b). Although *S2CoM* (an intuitive baseline approach) can reduce these discrepancies, it results in wrong predictions by the classifiers due to a complete loss of inter-entity information. To address the inter-entity distribution discrepancy problem, we propose a Convex Hull Adaptive Shift based multi-Entity action recognition method (CHASE). Serving as an additional normalization step, CHASE aims to accompany other single-entity backbones for enhanced multi-entity action recognition. Our main insight lies in the adaptive repositioning of skeleton sequences to mitigate inter-entity distribution gaps, thereby unbiasing the subsequent backbone and boosting its performance.
>
> ---
> **Q2:** *The sketch depicted in Figure 1 is a bit confusing and does not really illustrate what the authors aim to solve.  I would suggest the authors to provide a clear motivation example that leads to their method. The caption in Fig. 1 is rather poorly written.*
>
> We apologize for any confusion caused by Figure 1. We clarify the motivation example in Figure 1 that leads to our method as follows:
>
> 1. **What we aim to solve**: When using Vanilla (a common practice), the estimated distributions of joints from different entities show significant discrepancies, as shown in Figure 1(a). **These discrepancies can introduce bias into backbone models, leading to suboptimal optimization and poor recognition performance**, as depicted in the upper histogram of Figure 1(b). Though S2CoM (an intuitive baseline approach) can mitigate the discrepancies, **it makes the classifiers produce wrong predictions due to a complete loss of inter-entity information**. Therefore, this figure visualizes the problem we aim to solve.
> 2. **The effectiveness of CHASE**: Figure 1(a) and the lower histogram of Figure 1(b) highlight that CHASE effectively mitigates these discrepancies. Our method helps reduce bias in the subsequent classifiers, thereby enhancing their performance in the recognition task.
>
> In the revised version of the paper, we have updated Figure 1 and its caption to clearly illustrate the motivations and contributions mentioned above (see the pdf file in global response). The caption of Fig.1 is modified as follows:
>
> Figure 1: **Inter-entity distribution discrepancies in multi-entity action recognition task.** (a) We delineate three distinct settings: *Vanilla* (a common practice), *S2CoM* (an intuitive baseline approach), and *CHASE* (our proposed method). Column 2 illustrates spatiotemporal point clouds defined by the skeletons over $10^4$ sequences. Column 3-5 depict the projections of estimated distributions of these point clouds onto the x-y, z-x, and y-z planes. These projections reveal significant inter-entity distribution discrepancies when using *Vanilla*. (b) The discrepancies observed in *Vanilla* introduce bias into backbone models, leading to suboptimal optimization and poor performance. Although *S2CoM* can reduce these discrepancies, it makes the classifiers produce wrong predictions due to a complete loss of inter-entity information. With the lowest inter-entity discrepancy, our method unbiases the subsequent backbone to get the highest accuracy, underscoring its efficacy.

---

> ### Author Response · Authors · 2024-08-12
> **Looking Forward to Further Discussions**
>
> Dear Reviewer,
>
> Thanks again for your insightful comments on our paper.
>
> We have submitted the response to your comments and the global response with a PDF file. Please let us know if you have additional questions so that we can address them during the discussion period. We hope that you can consider rasing the score.
>
> Thank you

---

> > ### Comment · Reviewer_N2tt · 2024-08-13
> > **Answer**
> >
> > I am happy with the provided response and I have no further questions in regards to it. I believe this paper lies above the acceptance threshold.

---

> > > ### Author Response · Authors · 2024-08-13
> > > **Appreciation for Your Support and Positive Review**
> > >
> > > Dear Reviewer,
> > >
> > > Thank you for your positive feedback and for your confidence in our work. We appreciate your thoughtful review and are pleased that our responses met your expectations.
> > >
> > > Thank you again for your support and your constructive comments.
> > >
> > > Best regards,

---

### Author Rebuttal · Authors · 2024-08-06

We would like to express our sincere gratitude to all the reviewers for their time, insightful suggestions, and valuable comments. We deeply appreciate the positive recognition from the reviewers regarding our paper’s motivation (hFtj, 16iH, WotV), the elegance and simplicity of our methodology (N2tt, WotV), the rigorous mathematical derivation (N2tt), the extensive experiments demonstrating strong performance across a wide range of datasets and backbones (N2tt, 16iH, WotV), and the thorough implementation details provided for reproducibility (N2tt, hFtj). We address the common concerns raised by the reviewers below:

*1. Presentation of our motivations and contributions.*

We have revised the final paragraph of the Introduction section, Figure 1, and caption of Figure 1 to better articulate our motivations and contributions.

The contributions of this paper are three-fold:

- To the best of our knowledge, we are the first to investigate the issue of inter-entity distribution discrepancies in multi-entity action recognition. Our proposed method, Convex Hull Adaptive Shift for Multi-Entity Actions, effectively addresses this challenge. **Our main idea is to adaptively repositioning skeleton sequences to mitigate inter-entity distribution gaps, thereby unbiasing the subsequent backbones and boosting their performance.**
- **Serving as an additional normalization step for backbone models, CHASE consists of a learnable network and an auxiliary objective.** Specifically, the network is formulated by the Implicit Convex Hull Constrained Adaptive Shift, together with the parameterization of a lightweight Coefficient Learning Block, which learns sample-adaptive origin shifts within skeleton convex hull. Additionally, the Mini-batch Pair-wise Maximum Mean Discrepancy objective is proposed to guide the discrepancy minimization.
- Experiments on NTU Mutual 11, NTU Mutual 26, H2O, Assembly101, Collective Activity Dataset and Volleyball Dataset **consistently verify our proposed method by enhancing performance of single-entity backbones in multi-entity action recognition task**.

*2. Generalization of CHASE.*

- Compared with recent related works, **we have adopted more datasets with a wide range of types of entities and action categories**. As reported in Table 1 of our submission, we have verified CHASE across six diverse benchmarks, including person-to-person interactions, hand-to-object interactions, and group activities. Notably, we have evaluated CHASE on ASB101, which is the largest and most challenging benchmark for this task, featuring over 80,000 samples and 1,380 "verb+noun" categories, with absent object poses. (16iH Q8)
- We have evaluated CHASE using **both 3D joints and estimated 2D joints** as input. (16iH Q5)
- Experimental results demonstrate that **CHASE can be transferred without retraining the module if we align the input shapes of two datasets**. (hFtj Q4)

In addition to addressing the common concerns mentioned above, we have provided detailed responses to each specific question raised by the reviewers. We hope our responses will effectively address the reviewers’ concerns, and we look forward to engaging in comprehensive discussions in the coming days.

---

### Decision · Program_Chairs · 2024-09-25

**Decision:**

Accept (poster)

**Comment:**

3 of 4 of the reviewers found the paper's contributions significant with strong theoretical derivation for the technique and improved results across a number of benchmarks. The one negative reviewer found the motivation unclear and was not satisfied that model didn't support "transferr[ing] among different datasets without retraining the module". In the opinion of the AC these reasons don't qualify as reasons for rejection. Given this the AC recommends the paper be accepted to NeurIPs.